# Generating Molecules based on Latent Diffusion Model for Hierarchical Graph

## Abstract

Recently, generative models based on the diffusion process have emerged as a promising direction for automating the design of molecules. However, directly adding continuous Gaussian noise to discrete graphs leads to the problem of the final noisy data not conforming to the standard Gaussian distribution. Current graph diffusion models either corrupt discrete data through a transition matrix or relax the discrete data to continuous space for the diffusion process. These approaches not only require significant computation resources due to the inclusion of the bond type matrix but also cannot easily perform scalable conditional generation, such as adding cross-attention layers, due to the lack of embedding representations. In this paper, we introduce the Graph Latent Diffusion Model (GLDM), a novel variant of latent diffusion models that overcomes the mismatch problem of continuous diffusion space and discrete data space. Meanwhile, the latent diffusion framework avoids the issues of computational resource consumption and lack of embeddings for conditional generation faced by current graph diffusion models. However, it only utilizes graph-level embeddings for molecule generation, losing node-level and structural information. Therefore, we further extend the GLDM to the Latent Diffusion Model for Hierarchical Graph (HGLDM). By including node embeddings and subgraph embeddings that contain structural information, our model significantly reduces computation time compared to the current graph diffusion models. We evaluate our model on four benchmarks through the unconditional generation and conditional generation tasks, demonstrating its superior performance.

## 1 Introduction

Molecule generation is the process of creating novel molecular structures with desired properties and functions for various applications such as drug discovery, material engineering, and chemical synthesis (Chen et al., 2018). Recently, diffusion-based generative models have emerged as a promising direction for the automated design of molecules. These diffusion models are a class of probabilistic models that learn to generate realistic data by reversing a stochastic diffusion process that gradually transforms data into noise, which has achieved remarkable success in various image generation tasks such as image inpainting and image-to-text translation (Croitoru et al., 2023). However, most of the previous diffusion models rely on continuous Gaussian noise, which may lead to the final noisy data not conforming to the standard Gaussian distribution since the mismatch problem of continuous diffusion space and discrete data space (Niu et al., 2020).

Existing diffusion-based approaches for generating discrete graphs can be broadly classified into two types: one type employs transition matrices to iteratively modify discrete features as a way of controlling the forward corruption process (Austin et al., 2021). However, these approaches sacrifice the random exploration ability to ensure that the final noisy data conforms to the appropriate discrete category distribution. The other type relaxes discrete features to continuous features, but this type incurs a high memory cost due to the bond type matrices, limiting its applicability to large-scale datasets (Haefeli et al., 2022). In addition, due to the lack of embedding representation, these methods are difficult to extend, such as introducing conditional cross-attention layers.

Inspired by the achievements of latent diffusion (Rombach et al., 2022) in computer vision, we first introduce the Graph Latent Diffusion Model (GLDM), a novel variant of latent diffusion models for

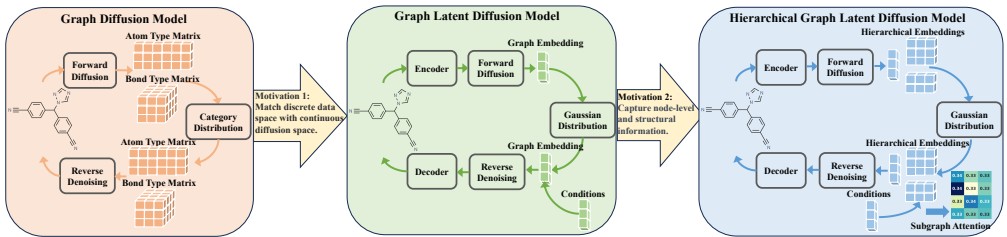

Figure 1: The motivations of the proposed methods. **Motivation 1:** We introduce GLDM to overcome the mismatch problem of continuous diffusion space and discrete data space. At the same time, this latent diffusion framework provides less compute consumption and scalable embeddings for conditional generation. **Motivation 2:** We propose HGLDM to capture node-level and structural information. This approach improves conditional generation by capturing the relationship between subgraphs and molecular properties.

molecule generation in continuous space, to overcome the mismatch problem of continuous diffusion space and discrete data space (see Figure 1, **Motivation 1**). Based on this framework, GLDM also addresses the issues of high computational cost and lack of latent embeddings for conditional generation that exist in other graph diffusion models. However, this method encounters the problem of insufficient node-level information and structural information when sampling exclusively in the graph-level latent space, leading to sub-optimal graph-level diffusion outcomes.

Therefore, we further propose the Latent Diffusion Model for Hierarchical Graph (HGLDM) in a hierarchical manner to involve more information on graphs (see Figure 1, **Motivation 2**). HGLDM extends GLDM to 3 hierarchical levels of latent space, namely node-level, subgraph-level (substructure-level), and graph-level (structure-level) latent spaces. We first employ the hierarchical autoencoder to encode hierarchical embeddings within molecules. Then, we propose a hierarchical diffusion model to generate hierarchical embeddings on continuous space. Based on the latent diffusion framework, we can achieve conditional molecule generation by using cross-attention to combine the desired molecular properties with the hierarchical embeddings. Our contributions are summarized as follows:

- We propose a Latent Diffusion Model for Hierarchical Graph (HGLDM) that leverages the idea from the latent diffusion framework by incorporating node-level and subgraph-level information into graph embeddings. This framework avoids the computational resource consumption that is typically associated with the diffusion process on bond type matrices, while still preserving hierarchical structure information.

- The proposed model's embedding representation provides a scalable solution for conditional molecule generation, allowing for the incorporation of desired molecular properties using cross-attention. This enables the HGLDM to generate diverse and high-quality molecular structures that meet various targets, while also possessing the ability to explore the relationship between subgraphs and molecular properties.

- We also evaluate the naive latent diffusion framework with graph-level embeddings, i.e., the graph latent diffusion model (GLDM). GLDM generates graph embeddings to overcome the mismatch problem of continuous diffusion space and discrete data space faced by traditional diffusion-based approaches. However, we observe that GLDM leads to sub-optimal graph-level diffusion outcomes.

We further evaluate our model on four benchmark datasets and compare our model with several state-of-the-art baselines on unconditional and conditional generation tasks, which demonstrate its superior performance.

## 2 PROBLEM DEFINITION

A molecular graph is represented by the tuple $G = (\mathbf{V}, \mathbf{E})$ with atom set $\mathbf{V} = \{v_1, \cdots, v_n\}$, bond type set $\mathbf{E} = \{e_{(v_i, v_j)} | v_i, v_j \in \mathbf{V}\}$. We denote by $n = |\mathbf{V}|$ and $m = |\mathbf{E}|$ the number of atoms and bonds in $G$, respectively. Each atom and bond has a corresponding categorical label, we denote by $\mathbf{X} = \{x_1, \cdots, x_n\}$ the atom types, by $e_{(v_i, v_j)}$ the bond type between atom $v_i$ and atom $v_j$. Given a set of training molecular graphs sampled from the data distribution $q(G)$, the goal of molecular generation is to learn the distribution of the observed set of molecular graphs $p(G)$. By sampling

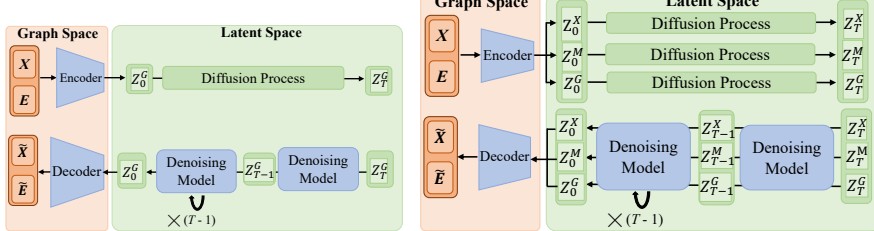

(a) Graph Latent Diffusion Model.  (b) Latent Diffusion Model for Hierarchical Graph.

Figure 2: Overall framework of GLDM and HGLDM. While GLDM (2a) encounters the problem of insufficient node-level and structural information when sampling in the graph-level latent space, HGLDM (2b) alleviates this problem by incorporating the node-level and subgraph-level information with a hierarchical design.

a graph $G_{new} \sim p(G)$, the new molecule can be generated. In this paper, we aim to solve the molecular generation problem by training the diffusion model based on training data.

## 3  DIFFUSION MODELS

The diffusion model (Sohl-Dickstein et al., 2015) is a type of generative model that consists of two Markov chains: a *forward diffusion process* and a *reverse denoising process*.

**Forward Diffusion Process.**  Given a data sample $\mathbf{z}_0 \sim q(\mathbf{z_0})$, the *forward diffusion process* $q(\mathbf{z}_{1:T}|\mathbf{z}_0) = \prod_{t=1}^{T} q(\mathbf{z}_t|\mathbf{z}_{t-1})$ generates a sequence of increasingly noisy latent variables $\mathbf{z}_{1:T} = \mathbf{z}_1, \mathbf{z}_2, \cdots, \mathbf{z}_T$ by gradually adding Guassian noise to corrupt the data sample $\mathbf{z}_0$. The forward diffusion process at time step $t$ is defined as $q(\mathbf{z}_t|\mathbf{z}_{t-1}) = \mathcal{N}(\mathbf{z}_t; \sqrt{1 - \beta_t}\mathbf{z}_{t-1}, \beta_t \mathbf{I})$, where the hyperparameter $\beta_t \in (0, 1)$ controls the amount of Gaussian noise $\mathcal{N}$ mixed with $\mathbf{z}_{t-1}$ at time step $t$. $\beta_{1:T}$ is determined by the noise schedule to ensure that the final latent variable $\mathbf{z}_T$ approximates a standard Gaussian noise, i.e., $\mathbf{z}_T \sim \mathcal{N}(\mathbf{0}, \mathbf{I})$.

**Reverse Denoising Process.**  The *reverse denoising process* aims to generate data sample $\mathbf{z}_0$ by denoising the noisy variables $\mathbf{z}_{T:1}$ toward target data distribution iteratively. However, since the *reverse denoising process* at each time step $q(\mathbf{z}_{t-1}|\mathbf{z}_t)$ is intractable, a parameterized Gaussian transitions $p_\theta(\mathbf{z}_{t-1}|\mathbf{z}_t)$ is designed to approximate the $q(\mathbf{z}_{t-1}|\mathbf{z}_t)$ at each time step as $p_\theta(\mathbf{z}_{t-1}|\mathbf{z}_t) = \mathcal{N}(\mathbf{z}_{t-1}; \boldsymbol{\mu}_\theta(\mathbf{z}_t, t), \sigma^2 \mathbf{I})$, where the means $\boldsymbol{\mu}_\theta$ denotes a neural network with learnable parameters $\theta$, and the variances $\sigma^2$ are predefined. The parameterized *reverse denoising process* is then defined as $p_\theta(\mathbf{z}_{0:T}) = p(\mathbf{z}_T) \prod_{t=1}^{T} p_\theta(\mathbf{z}_{t-1}|\mathbf{z}_t)$, where $p(\mathbf{z}_T)$ is a standard Gaussian distribution.

## 4  LATENT DIFFUSION MODEL FOR HIERARCHICAL GRAPH

In this section, we first introduce the Latent Diffusion Model for Hierarchical Graph (HGLDM), as illustrated in Figure 2b. We introduce the hierarchical autoencoder at the beginning, which maps discrete molecular features to continuous hierarchical latent space in Section 4.1. Next, we elaborate on the hierarchical diffusion model in Section 4.2, which considers information interaction across atoms, subgraphs, and the whole graph in the latent space. Then, we describe the training and sampling processes of HGLDM in Section 4.3. In Section 4.4, we introduce a conditional generation method. Finally, Section 4.5 describes our initial idea of applying the naive latent diffusion framework for graphs named as graph latent diffusion model (GLDM) formulated in Figure 2a.

### 4.1  HIERARCHICAL AUTOENCODER

Due to the memory-intensive and time-consuming nature of directly performing the diffusion process on the bond type matrix, most diffusion-based graph generation methods encounter challenges in generating large-scale molecules. To address this, we employ subgraph embeddings as substitutes for bond embeddings which incorporate subgraphs as local structural information. To comprehensively encode the molecular information, we also introduce fine-grained atom embeddings and coarse-grained graph embedding, facilitating a hierarchical encoding of the molecule.

#### 4.1.1  ENCODER AND DECODER

We denote by $\mathbf{z}^X \in \mathbb{R}^{n \times d_x}$ the atom embeddings, by $\mathbf{z}^M \in \mathbb{R}^{n \times d_m}$ the subgraph embeddings representing the subgraphs each atom belongs to, and by $\mathbf{z}^G \in \mathbb{R}^{1 \times d_g}$ the graph embedding. The

Encoder $\mathcal{E}_\phi$ and Decoder $\mathcal{D}_\psi$ are formulated as follows:

$$q_\phi(\mathbf{z}^X, \mathbf{z}^M, \mathbf{z}^G | \mathbf{X}, \mathbf{E}) = \mathcal{N}(\mathcal{E}_\phi(\mathbf{X}, \mathbf{E}), \sigma \boldsymbol{I}),$$

$$p_\psi(\mathbf{X}, \mathbf{E} | \mathbf{z}^X, \mathbf{z}^M, \mathbf{z}^G) = \prod_{i=1}^n p_\psi(x_i | \mathbf{z}^X, \mathbf{z}^M, \mathbf{z}^G) \prod_{i=1}^n \prod_{j=1, j \neq i}^n p_\psi(e_{(v_i, v_j)} | \mathbf{z}^X, \mathbf{z}^M, \mathbf{z}^G), \quad (1)$$

where $\phi$ and $\psi$ are the trainable parameters of the Encoder $\mathcal{E}_\phi$ and Decoder $\mathcal{D}_\psi$, respectively. In this paper, we evaluate the state-of-the-art VAE-based method, PS-VAE (Kong et al., 2022), for implementing the molecular autoencoder. In PS-VAE, the authors first propose a *Principal Sub-graph* method to extract subgraphs, then they utilize GIN (Xu et al., 2019) with edge features as the Encoder to convert molecular graph $G$ into hierarchical latent embeddings. The Decoder, an autoregressive model implemented with a single GRU (Cho et al., 2014) layer, converts latent embeddings into a sequence of graph fragments. Lastly, the link predictor predicts connections between these fragments to construct the molecule. However, our proposed framework is not limited to this specific model. Alternative models such as JTVAE (Jin et al., 2018), HierVAE (Jin et al., 2020), and MiCaM (Geng et al., 2022) can also be considered for this purpose.

### 4.1.2 TRAINING LOSS

The autoencoder is trained by minimizing the reconstruction loss and Kullback-Leibler (KL) loss:

$$\begin{aligned} \mathcal{L}_{\text{AE}} &= \mathcal{L}_{\text{Rec}} + \gamma \mathcal{L}_{\text{KL}} \\ &= -\mathbb{E}_{q_\phi(\mathbf{z}^X, \mathbf{z}^M, \mathbf{z}^G | \mathbf{X}, \mathbf{E})} p_\psi(\mathbf{X}, \mathbf{E} | \mathbf{z}^X, \mathbf{z}^M, \mathbf{z}^G) \\ &\quad + \gamma D_{KL}(q_\phi(\mathbf{z}^X, \mathbf{z}^M, \mathbf{z}^G | \mathbf{X}, \mathbf{E}) || p(\mathbf{z}^X, \mathbf{z}^M, \mathbf{z}^G)), \end{aligned} \quad (2)$$

where $\gamma$ is a hyperparameter used to set the weight of the KL loss. We use the KL loss to align the latent space features with the standard Gaussian distribution $p(\mathbf{z}^X, \mathbf{z}^M, \mathbf{z}^G)$. The training loss $\mathcal{L}_{\text{AE}}$ balances the reconstruction error and the KL-divergence between the prior and the posterior distributions of the latent embeddings, providing a lower bound on the log-likelihood of the data (Kingma & Welling, 2014). This results in a compact and meaningful latent space for generating appropriate hierarchical embeddings in Hierarchical Diffusion. To differentiate different time steps in the diffusion model, we denote $\mathbf{z}^X$, $\mathbf{z}^M$, and $\mathbf{z}^G$ as $\mathbf{z}_0^X$, $\mathbf{z}_0^M$, and $\mathbf{z}_0^G$ in the subsequent sections.

### 4.2 HIERARCHICAL DIFFUSION MODEL

Given the latent hierarchical embeddings $\mathbf{z}_0^X$, $\mathbf{z}_0^M$, and $\mathbf{z}_0^G$ encoded by the Molecular Encoder $\mathcal{E}_\phi$, the hierarchical diffusion model generates noisy latent variables $\mathbf{z}_t^X$, $\mathbf{z}_t^M$, and $\mathbf{z}_t^G$ during the *forward diffusion process*. During the *reverse denoising process*, we sample the initiate embeddings $\mathbf{z}_T^X$, $\mathbf{z}_T^M$, and $\mathbf{z}_T^G$ from the standard Gaussian distribution. We propose a Hierarchical Denoising Neural Network to predict the noises $\hat{\epsilon}_t^X$, $\hat{\epsilon}_t^M$, and $\hat{\epsilon}_t^G$ at each reverse denoise step.

### 4.2.1 HIERARCHICAL DENOISING NEURAL NETWORK

Figure 3a illustrates the architecture of the Hierarchical Denoising Neural Network. First, we apply a Multi-Layer Perceptron (MLP) to each of these embeddings to obtain their corresponding latent representations $\mathbf{h}_t^X$, $\mathbf{h}_t^M$, and $\mathbf{h}_t^G$, respectively:

$$\mathbf{h}_t^X = \text{MLP}(\mathbf{z}_t^X), \mathbf{h}_t^M = \text{MLP}(\mathbf{z}_t^M), \mathbf{h}_t^G = \text{MLP}(\mathbf{z}_t^G). \quad (3)$$

Next, we input these latent representations into the Hierarchical Block.

**Hierarchical Block.** The Hierarchical Block is a module that processes and combines the latent representations in a hierarchical manner, taking into account the relationships between atomic-level representations, subgraph-level representations, and global graph-level representations, its architecture is shown in Figure 3b. The update process for each level is as follows:

1. Update atom embeddings: For atom and subgraph embeddings, we apply an MLP transformation to each level and then sum the resulting embeddings to update the atom embeddings:

$$\mathbf{h}_t^{X'} = \text{MLP}(\mathbf{h}_t^X) + \text{MLP}(\mathbf{h}_t^M). \quad (4)$$

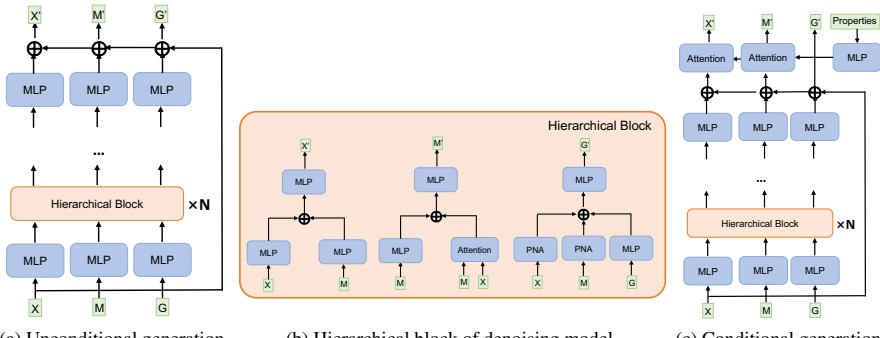

Figure 3: Hierarchical Denoising Neural Network.

2. Update subgraph embeddings: First, we compute the attention scores between subgraph embeddings and atom embeddings using dot product. Next, we update the subgraph embeddings by multiplying the attention scores with the MLP-transformed atom embeddings. Finally, we incorporate the original subgraph-level information into the updated subgraph embeddings:

$$\mathbf{h}_{t,i}^{M'} = \sum_j \text{softmax}\left(\mathbf{h}_{t,i}^M \mathbf{h}_{t,j}^{X}{}^\top\right) \text{MLP}(\mathbf{h}_{t,j}^X) + \mathbf{h}_{t,i}^M. \tag{5}$$

3. Update graph embeddings: In the process of updating graph embeddings, we first use Principal Neighbourhood Aggregation (PNA) pooling (Corso et al., 2020) to obtain atom-level representation and subgraph-level representation. Then, atom-level representation and subgraph-level representation are combined with the MLP-transformed graph embedding:

$$\mathbf{h}_t^{G'} = \text{PNA}(\mathbf{h}_t^X) + \text{PNA}(\mathbf{h}_t^M) + \text{MLP}(\mathbf{h}_t^G). \tag{6}$$

In the hierarchical block, we use different MLP functions to indicate that these transformations have separate parameters and are not shared between different operations. These updated embeddings, $\mathbf{h}_t^{X'}$, $\mathbf{h}_t^{M'}$, and $\mathbf{h}_t^{G'}$ are then used as the input for the next layer of the Hierarchical Block.

Subsequently, we use another set of MLPs to predict the atomic-level noise, subgraph-level noise, and graph-level noise from the Hierarchical Block's output. We then add these predictions as residual connections to the original input embeddings:

$$\hat{\epsilon}_t^X = \mathbf{z}_t^X + \text{MLP}(\mathbf{h}_t^{X'}), \hat{\epsilon}_t^M = \mathbf{z}_t^M + \text{MLP}(\mathbf{h}_t^{M'}), \hat{\epsilon}_t^G = \mathbf{z}_t^G + \text{MLP}(\mathbf{h}_t^{G'}). \tag{7}$$

These predicted noise values, $\hat{\epsilon}_t^X$, $\hat{\epsilon}_t^M$, and $\hat{\epsilon}_t^G$, are used to update the embeddings during the reverse denoising process. By incorporating the hierarchical structure into the denoising model, our method effectively captures the relationships between different levels of molecular features and generates more accurate and diverse molecular structures.

### 4.3 TRAINING AND SAMPLING

In this section, we describe the training and sampling procedures of the Latent Diffusion Model for Hierarchical Graph (HGLDM) in detail.

**Training.** The training process of the HGLDM consists of two stages, as described in Algorithm 1 in the Appendix A. In the first stage, we train the molecular autoencoder, which includes the molecular encoder $\mathcal{E}_\phi$ and molecular decoder $\mathcal{D}_\psi$. We optimize the objective function $\mathcal{L}_{AE}$, which consists of the reconstruction loss $L_{Rec}$ and the Kullback-Leibler (KL) divergence loss $\mathcal{L}_{KL}$, weighted by a hyperparameter $\gamma$. We continue optimizing until the parameters $\phi$ and $\psi$ have converged.

In the second stage, we train the hierarchical denoising neural network $\epsilon_\theta$ together with the encoder $\mathcal{E}_\phi$. We optimize the objective function $\mathcal{L}$, which is the expected squared error between the denoising neural network's output and the original noise, weighted by the reweighting terms $w(t)$. We continue optimizing until the parameters $\theta$ and $\phi$ have converged.

**Sampling.** The sampling process of the HGLDM, as described in Algorithm 2 in Appendix A, involves generating new molecules using the trained model. To generate a new molecule, we first

sample the initial latent variables $\mathbf{z}_T^{\mathbf{X}}, \mathbf{z}_T^{\mathbf{M}}, \mathbf{z}_T^{\mathbf{G}}$ from a standard Gaussian distribution. We then perform a reverse diffusion process by iterating through the diffusion time steps $T$. In each step, we sample a noise vector $\epsilon$ and update the latent variables according to the hierarchical denoising neural network $\epsilon_\theta$ and the noise schedule. Finally, we decode the latent variables $\mathbf{z}_0^{\mathbf{X}}, \mathbf{z}_0^{\mathbf{M}}, \mathbf{z}_0^{\mathbf{G}}$ using the molecular decoder $\mathcal{D}_\psi$ to obtain the generated molecule's atom features $\mathbf{X}$ and edge features $\mathbf{E}$.

## 4.4 CONDITIONAL GENERATION

In this section, we introduce a conditional generation approach that leverages cross attention to incorporate molecular properties as conditions for generating molecules with specific properties, the architecture is shown in Figure 3c.

Given the updated atomic embeddings $\mathbf{h}_t^X$, subgraph embeddings $\mathbf{h}_t^M$, and graph embedding $\mathbf{h}_t^G$ after the multi-layer hierarchical blocks, we aim to condition the generation process on the specific molecular properties. Let $\mathbf{s} = \{s_1, s_2, \cdots\}$ denote the molecular property embedding, where each dimension represents a specific molecular property. We first process atomic embeddings, subgraph embeddings, graph embeddings, and molecular property embedding $\mathbf{s}$ through the MLP to obtain query, key, and value representations:

$$Q^X = \mathrm{MLP}(\mathbf{h}_t^X), Q^M = \mathrm{MLP}(\mathbf{h}_t^M), Q^G = \mathrm{MLP}(\mathbf{h}_t^G), K = \mathrm{MLP}(\mathbf{s}), V = \mathrm{MLP}(\mathbf{s}). \quad (8)$$

The cross-attention mechanism consists of two main steps: computing the attention scores and updating the embeddings using the attention scores and values. First, the attention scores are calculated using the dot product between the query and key representations, followed by a softmax operation for normalization:

$$\alpha_{ij}^X = \mathrm{softmax}(Q_i^X K_j^T / \sqrt{d_k}), \alpha_{ij}^M = \mathrm{softmax}(Q_i^M K_j^T / \sqrt{d_k}), \alpha_j^G = \mathrm{softmax}(Q^G K_j^T / \sqrt{d_k}),$$

where $d_k$ is the key dimension. The attention scores $\alpha_{ij}$ capture the relevance between the atomic (subgraph) embeddings and desired molecular properties. Next, the updated embeddings are obtained by computing the weighted sum of the value representations using the attention scores:

$$\mathbf{h'}_t^X = \sum_j \alpha_{ij}^X V_j, \mathbf{h'}_t^M = \sum_j \alpha_{ij}^M V_j, \mathbf{h'}_t^G = \sum_j \alpha_j^G V_j, \quad (9)$$

The updated atomic embeddings $\mathbf{h'}_t^X$, subgraph embeddings $\mathbf{h'}_t^M$, and graph embeddings $\mathbf{h'}_t^G$ now contain information about the specific molecular properties, which guides the generation process towards molecules with the desired properties.

## 4.5 GRAPH LATENT DIFFUSION MODEL

As illustrated in Figure 2a, for GLDM, we first train an autoencoder, such as the classic VAE or the aforementioned hierarchical VAE models. These models are capable of obtaining node-level embeddings $\mathbf{z}_0^X$ through the encoder and then generating graph-level embeddings $\mathbf{z}_0^G$, via a pooling operation. The decoder then reconstructs the discrete molecular data from the graph-level embeddings. Subsequently, we train a diffusion model to explore the generative capabilities of various graph-level embeddings. In this process, we employ the MLPs as the denoising model. Furthermore, we achieve conditional generation by performing cross-attention between the latent graph embeddings and the given conditional embeddings.

## 5 EXPERIMENTS

In this section, we evaluate the performance of our proposed Latent Diffusion Model for Hierarchical Graph (HGLDM) on molecule generation tasks using benchmark datasets, state-of-the-art baselines, and various evaluation metrics. We conduct experiments on both unconditional and conditional molecule generation tasks, analyzing the results and computational efficiency of our model. The experimental setups are detailed in Appendix B.

### 5.1 UNCONDITIONAL GENERATION

**Computational Efficiency.** We compare the computational efficiency of GLDM and HGLDM with the baseline methods by measuring the training and sampling times on the GuacaMol dataset. The experimental results in Table 1 demonstrate that GLDM and HGLDM significantly outperform diffusion-based methods GDSS and DiGress in terms of computational efficiency, being approximately 3.6 to 4.3 times faster than DiGress during training and 83.5 to 123.4 times

faster during sampling on the GuacaMol dataset. This highlights the effectiveness of our models for large-scale molecule generation tasks, where efficiency is a crucial factor to consider.

**Quantitative Results.** Table 2 presents our experimental results of the unconditional molecule generation tasks. The results demonstrate that our proposed methods consistently achieve superior or competitive performance across the QM9, ZINC250K, GuacaMol and MOSES datasets. Compared to PS-VAE, our models demonstrate superior uniqueness, novelty, diversity, FCD, and NSPDK MMD. Furthermore, they significantly surpass diffusion-based methods regarding validity, novelty and diversity, showcasing their enhanced performance. In addition, compared to diffusion-based methods, our approaches achieve comparable unique results with fewer parameters and less time consumption. This highlights the advantages of the graph latent diffusion framework in generating more innovative and valid molecular structures. Although the proposed method does not achieve the best results in FCD and NSPDK MMD, this is mainly limited by the VAE backbone model, and we have improved upon PSVAE in terms of FCD and NSPDK across all the datasets. Furthermore, unlike general graph generation, which focuses more on learning the distribution of structures and features, comparing feature similarity in molecular generation without considering molecular properties lacks practical value.

Table 1: Training and sampling speed for our models and the baselines on GuacaMol.

| Methods | # of Params | Training (h) | Sampling (h) |
|---|---|---|---|
| PS-VAE | 637K | 12.58 | 0.17 |
| GDSS | 76.5K | 63.22 | 8.32 |
| DiGress | 4.6M | 109.00 | 21.14 |
| GLDM | 549K | 24.11 | 0.17 |
| HGLDM | 2.0M | 25.17 | 0.25 |

Table 2: Results of unconditional generation on QM9, ZINC250K, GuacaMol and MOSES.

| Datasets | Methods | Valid ↑ | Unique↑ | Novelty↑ | Diversity↑ | FCD↓ | NSPDK MMD↓ |
|---|---|---|---|---|---|---|---|
| QM9 | GDSS | $0.957 \pm 0.000$ | $0.982 \pm 0.003$ | $0.988 \pm 0.001$ | $\mathbf{0.925 \pm 0.000}$ | $2.959 \pm 0.040$ | $0.003 \pm 0.000$ |
| | DiGress | $0.992 \pm 0.003$ | $0.960 \pm 0.001$ | $0.391 \pm 0.001$ | $0.920 \pm 0.000$ | $\mathbf{2.123 \pm 0.033}$ | $\mathbf{0.001 \pm 0.000}$ |
| | PS-VAE | $\mathbf{1.000 \pm 0.000}$ | $0.981 \pm 0.002$ | $0.996 \pm 0.000$ | $0.881 \pm 0.000$ | $16.877 \pm 0.059$ | $0.059 \pm 0.000$ |
| | GLDM | $\mathbf{1.000 \pm 0.000}$ | $0.982 \pm 0.001$ | $0.996 \pm 0.000$ | $0.884 \pm 0.000$ | $14.829 \pm 0.169$ | $0.050 \pm 0.001$ |
| | HGLDM | $\mathbf{1.000 \pm 0.000}$ | $\mathbf{0.985 \pm 0.002}$ | $\mathbf{0.997 \pm 0.000}$ | $0.884 \pm 0.000$ | $14.576 \pm 0.183$ | $0.047 \pm 0.001$ |
| ZINC250K | GDSS | $\mathbf{1.000 \pm 0.000}$ | $0.997 \pm 0.001$ | $\mathbf{1.000 \pm 0.000}$ | $0.902 \pm 0.000$ | $16.086 \pm 0.071$ | $\mathbf{0.018 \pm 0.000}$ |
| | DiGress | $0.565 \pm 0.005$ | $\mathbf{1.000 \pm 0.000}$ | $\mathbf{1.000 \pm 0.000}$ | $0.882 \pm 0.000$ | $\mathbf{13.042 \pm 0.164}$ | $0.031 \pm 0.001$ |
| | PS-VAE | $\mathbf{1.000 \pm 0.000}$ | $0.993 \pm 0.001$ | $\mathbf{1.000 \pm 0.000}$ | $0.912 \pm 0.000$ | $20.386 \pm 0.061$ | $0.085 \pm 0.001$ |
| | GLDM | $\mathbf{1.000 \pm 0.000}$ | $0.994 \pm 0.000$ | $\mathbf{1.000 \pm 0.000}$ | $0.913 \pm 0.000$ | $20.444 \pm 0.088$ | $0.086 \pm 0.001$ |
| | HGLDM | $\mathbf{1.000 \pm 0.000}$ | $0.997 \pm 0.001$ | $\mathbf{1.000 \pm 0.000}$ | $\mathbf{0.914 \pm 0.000}$ | $19.913 \pm 0.091$ | $0.084 \pm 0.000$ |
| Guacamol | GDSS | $\mathbf{1.000 \pm 0.000}$ | $0.986 \pm 0.001$ | $0.996 \pm 0.001$ | $0.892 \pm 0.000$ | $40.291 \pm 0.072$ | $0.058 \pm 0.000$ |
| | DiGress | $0.875 \pm 0.005$ | $\mathbf{1.000 \pm 0.000}$ | $\mathbf{0.999 \pm 0.001}$ | $0.904 \pm 0.000$ | $\mathbf{12.069 \pm 0.051}$ | $\mathbf{0.018 \pm 0.000}$ |
| | PS-VAE | $\mathbf{1.000 \pm 0.000}$ | $0.998 \pm 0.000$ | $0.998 \pm 0.000$ | $\mathbf{0.905 \pm 0.000}$ | $24.105 \pm 0.082$ | $0.090 \pm 0.000$ |
| | GLDM | $\mathbf{1.000 \pm 0.000}$ | $0.998 \pm 0.000$ | $0.998 \pm 0.000$ | $0.904 \pm 0.000$ | $23.879 \pm 0.041$ | $0.095 \pm 0.000$ |
| | HGLDM | $\mathbf{1.000 \pm 0.000}$ | $0.999 \pm 0.001$ | $\mathbf{0.999 \pm 0.000}$ | $\mathbf{0.905 \pm 0.000}$ | $23.845 \pm 0.098$ | $0.095 \pm 0.001$ |
| MOSES | GDSS | $\mathbf{1.000 \pm 0.000}$ | $0.994 \pm 0.003$ | $0.999 \pm 0.000$ | $0.899 \pm 0.000$ | $21.265 \pm 0.249$ | $0.037 \pm 0.005$ |
| | DiGress | $0.858 \pm 0.005$ | $\mathbf{1.000 \pm 0.000}$ | $0.996 \pm 0.001$ | $0.886 \pm 0.000$ | $\mathbf{9.228 \pm 0.081}$ | $\mathbf{0.010 \pm 0.000}$ |
| | PS-VAE | $\mathbf{1.000 \pm 0.000}$ | $0.999 \pm 0.000$ | $\mathbf{1.000 \pm 0.000}$ | $0.905 \pm 0.000$ | $26.401 \pm 0.078$ | $0.079 \pm 0.000$ |
| | GLDM | $\mathbf{1.000 \pm 0.000}$ | $0.998 \pm 0.000$ | $\mathbf{1.000 \pm 0.000}$ | $0.905 \pm 0.000$ | $26.365 \pm 0.095$ | $0.077 \pm 0.001$ |
| | HGLDM | $\mathbf{1.000 \pm 0.000}$ | $0.999 \pm 0.000$ | $\mathbf{1.000 \pm 0.000}$ | $\mathbf{0.906 \pm 0.000}$ | $25.815 \pm 0.053$ | $0.072 \pm 0.000$ |

**Ablation Study.** As shown in Table 2, comparing PS-VAE, GLDM, and HGLDM, both GLDM and HGLDM outperform PS-VAE with higher uniqueness and novelty scores, showcasing their effectiveness in generating diverse and novel molecular structures. Additionally, compared to GLDM, the incorporation of atom embeddings and subgraph embeddings in HGLDM enhances its ability to capture complex structural information, resulting in more innovative molecular structures.

## 5.2 INTERPOLATION OF LATENT EMBEDDINGS

To showcase the improvements offered by the latent diffusion framework over traditional diffusion models (Jo et al., 2022a; Vignac et al., 2023) in molecular generation, we conduct an interpolation experiment on latent embeddings. We select 100 molecules with the highest QED (Quantitative Estimate of Drug-likeness) (Bickerton et al., 2012) and 100 molecules with the highest PlogP (Penalized logP) (Kusner et al., 2017) from the ZINC250K dataset. Utilizing this sampled dataset, we train the PSVAE and HGLDM models and carry out three sets of experiments: QED Interpolation, PlogP Interpolation, and Mixed Interpolation. QED Interpolation involves using only latent embeddings with high QED values, while PlogP Interpolation uses only those with high PlogP values. Mixed interpolation combines latent embeddings with high QED and high PlogP values. In each

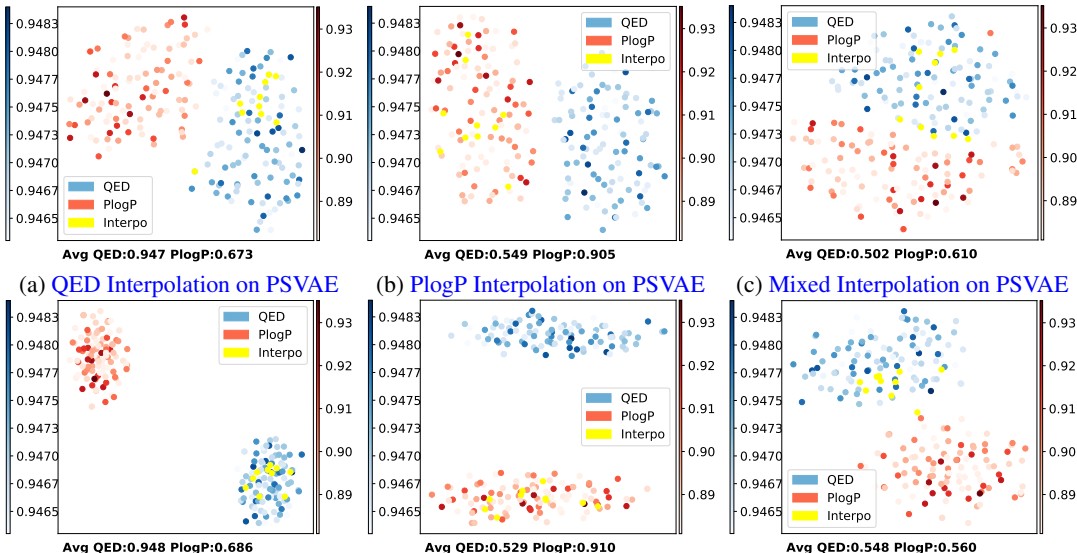

(a) QED Interpolation on PSVAE  (b) PlogP Interpolation on PSVAE  (c) Mixed Interpolation on PSVAE

(d) QED Interpolation on HGLDM  (e) PlogP Interpolation on HGLDM  (f) Mixed Interpolation on HGLDM

Figure 4: The results of interpolating latent embeddings of the PSVAE and HGLDM. We interpolate latent embeddings with high QED values (i.e., (a) and (d)), latent embeddings with high PlogP values (i.e., (b) and (e)), and mixed latent embeddings with high QED and PlogP attribute values (i.e., (c) and (f)).

interpolation, we randomly sample two latent embeddings, $\mathbf{z}^{G_1}$ and $\mathbf{z}^{G_2}$, and generate the interpolated embeddings $G'$ through $\mathbf{z}^{G'} = \lambda \mathbf{z}^{G_1} + (1 - \lambda)\mathbf{z}^{G_2}$, where $\lambda \in (0, 1)$. In QED Interpolation and PlogP Interpolation, we set the expected minimum QED to be 0.94 and the minimum PlogP to 0.89 and output 10 interpolated embeddings that first reached the expected minimum values. In the Mixed interpolation experiment, we randomly interpolate ten embeddings.

We employ t-SNE (Van der Maaten & Hinton, 2008) to visualize the corresponding latent embeddings and interpolated embeddings in Figure 4. Additionally, we annotate the average property values of the molecules decoded from the interpolated embeddings. From Figure 4, we can draw two key conclusions. Firstly, the clear boundaries between embeddings of different properties in the scatter plot, along with the high property values retained by the interpolated embeddings, suggest that latent embeddings can effectively distinguish between molecules with varying property values. This provides a solid foundation for further conditional generation using these latent embeddings. Secondly, latent embeddings trained by HGLDM capture more distinct features of different property values compared to PS-VAE latent embeddings, as evidenced by the farther distances between HGLDM latent embeddings with different property values and the higher property values of molecules decoded from HGLDM latent embeddings interpolation. This highlights the enhancement of the latent diffusion framework over VAE-based models.

## 5.3 CONDITIONAL GENERATION

To evaluate the conditional generation capabilities of HGLDM, we conduct an experiment on a set of 100 molecules sampled from the test dataset of ZINC250K and calculate their QED (Quantitative Estimate of Drug-likeness) (Bickerton et al., 2012), SA (Synthetic Accessibility) (Ertl & Schuffenhauer, 2009), and PlogP (Penalized logP) (Kusner et al., 2017) values as the conditioning values during training. The ablation study of the conditional generation can be found in Appendix C.1.

**Quantitative Results.** For GDSS, we employ MOOD (Lee et al., 2023) which uses classifier guidance based on GDSS, as its conditional generation framework. For DiGress, we concatenated the molecular property values to the graph-level features as additional features. The performance of each model is assessed by comparing the mean absolute error (MAE) between the target properties and the properties of the generated molecules, as shown in Table 3. The results show that both HGLDM and GLDM, which employ cross-attention mechanisms, outperform DiGress on all property generation tasks and outperform GDSS on single property generation tasks. This demonstrates the advantage of the latent diffusion framework in capturing the relationships between latent molec-

Table 3: MAE results of conditional generation on ZINC250K, the values in parentheses represent the standard deviation of the absolute error.

| Models | Task | QED | SA | PlogP | QED&SA&PlogP |
|---|---|---|---|---|---|
| GDSS | Unconditional | 0.437(0.006) | 0.058(0.025) | 0.114(0.044) | **0.176(0.027)** |
| | Conditional | 0.435(0.009) | 0.033(0.007) | 0.077(0.020) | 0.182(0.012) |
| DiGress | Unconditional | 0.859(0.009) | 0.679(0.010) | 0.788(0.013) | 0.775(0.010) |
| | Conditional | 0.531(0.018) | 0.152(0.011) | 0.173(0.024) | 0.207(0.019) |
| GLDM | Unconditional | 0.414(0.005) | 0.065(0.008) | 0.099(0.012) | 0.191((0.008) |
| | Conditional | 0.404(0.005) | 0.063(0.010) | 0.094(0.011) | 0.187(0.009) |
| HGLDM | Unconditional | 0.415(0.005) | 0.063(0.013) | 0.085(0.010) | 0.188(0.009) |
| | Conditional | **0.348(0.018)** | **0.021(0.010)** | **0.068(0.004)** | 0.185(0.008) |

ular embeddings and properties. In multi-properties generation, although GDSS performs the best, its conditional generation results are slightly worse compared to unconditional generation. Additionally, GDSS exhibits significant variance in the unconditional generation, resulting in unstable generated results. We also note that DiGress generates many invalid molecules during the conditional generation process, possibly due to the limitations of concatenating molecular properties as conditions, which cannot be effectively learned by node features and chemical bond features in the denoising network. Comparing HGLDM to GLDM, we observe that different from the slight improvement in unconditional generation task, HGLDM can significantly improve the performance over GLDM in conditional generation task. This is because the properties of molecules are often determined by distinctive functional groups (subgraphs). The hierarchical embeddings in HGLDM can effectively learn the influence of molecular properties on subgraphs, enabling the model to generate molecules with desired properties more accurately. Additionally, HGLDM achieves better results than both GLDM and DiGress, even in the unconditional setting. This highlights the effectiveness of the hierarchical structure in capturing more information about molecular properties.

## 6 RELATED WORK

**Diffusion-based Molecule Generation.** In recent years, generative models based on diffusion processes have emerged as a promising direction for molecular automated design (Liu et al., 2023; Lee et al., 2023). Most of the previous diffusion models have the mismatch problem of continuous diffusion space and discrete data space (Niu et al., 2020). To address this limitation, recent works (Haefeli et al., 2022; Chen et al., 2022; Vignac et al., 2023; Luo et al., 2022) have proposed to utilize discrete noise instead of Gaussian noise. Besides those methods on graph structures, there are also several works that define diffusion models in 3D atomic positions (Trippe et al., 2022; Hoogeboom et al., 2022; Bao et al., 2022; Huang et al., 2022; Qiang et al., 2023; Xu et al., 2023). To the best of our knowledge, our proposed method is the first work that incorporates the hierarchical structure information into the design of discrete denoising diffusion model for the molecule generation.

**Hierarchical Graph Learning.** Hierarchical graph learning extends GNNs by incorporating multiple levels of abstraction and resolution in the graph, which can capture both local and global information about the graph (Gao et al., 2023; Defferrard et al., 2016; Ying et al., 2018; Gao & Ji, 2019). It was only recently that the power of graph hierarchy in graph generation has been explored (Jin et al., 2020; Kuznetsov & Polykovskiy, 2021; Qu & Zou, 2022; Karami & Luo, 2023). Despite some differences, all of these methods learn the hierarchy for graph generation using VAEs, GANs, or normalizing flows. In contrast, our hierarchy is constructed to incorporate the diffusion model on graphs in the discrete space.

## 7 CONCLUSION AND FUTURE WORK

In this work, we propose the Graph Latent Diffusion Model (GLDM) and the Latent Diffusion Model for Hierarchical Graph (HGLDM) as novel approaches for molecule generation. These models effectively address the challenges of time consumption and lack of latent embedding for conditional generation faced by previous diffusion-based approaches. Specifically, compared to traditional approaches, HGLDM learns both the structural and molecular property information of molecules by incorporating hierarchical embeddings. Our experiments on benchmark datasets demonstrate the superior performance of HGLDM in both unconditional and conditional molecule generation, showcasing its ability to generate diverse and high-quality molecules that meet various objectives. As future work, our model can be further extended to incorporate language models for multimodal conditional generation, such as using MoleculeSTM (Liu et al., 2022) to describe the desired molecular properties and guide the HGLDM for conditional generation.

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

# A    DETAILS OF TRAINING AND SAMPLING

## A.1    TRAINING AND SAMPLING FOR DIFFUSION MODEL

By assuming $\bar{\alpha}_t = \prod_{s=1}^{t}(1 - \beta_s)$, the noisy latent variable $\mathbf{z}_t$ at time step $t$ can be formulated via re-parameterization trick (Kingma & Welling, 2014) as $q(\mathbf{z}_t|\mathbf{z}_0) = \mathcal{N}(\mathbf{z}_t; \sqrt{\bar{\alpha}_t}\mathbf{z}_0, (1 - \bar{\alpha}_t)\boldsymbol{I})$. In this manner, we could generate noisy latent variable $\mathbf{z}_t = \sqrt{\bar{\alpha}_t}\mathbf{z}_0 + (\sqrt{1 - \bar{\alpha}_t})\boldsymbol{\epsilon}$ directly from $\mathbf{z}_0$ by mixing it with Gaussian noise $\boldsymbol{\epsilon} \sim \mathcal{N}(\mathbf{0}, \boldsymbol{I})$ instead of iterating $q(\mathbf{z}_t|\mathbf{z}_{t-1})$ $t$ times. Then, the following objective (Song & Ermon, 2019) is adopted to train the parameterized neural network:

$$\mathcal{L} = \mathrm{E}_{\mathbf{z}_0, \boldsymbol{\epsilon} \sim \mathcal{N}(0, \boldsymbol{I}), t}[w(t)||\boldsymbol{\epsilon} - \boldsymbol{\epsilon}_\theta(\mathbf{z}_t, t)||^2], \tag{10}$$

where $w(t) = \frac{\beta_t^2}{2\sigma^2(1-\beta_t)(1-\bar{\alpha}_t)}$ are reweighting terms. With trained neural network $\boldsymbol{\epsilon}_\theta$, we could predict the Gaussian noise $\boldsymbol{\epsilon}$ at time step $t$. The sampling process can be conducted as $\mathbf{z}_{t-1} = \frac{1}{\sqrt{1-\beta_t}}(\mathbf{z}_t - \frac{\beta_t}{\sqrt{1-\bar{\alpha}_t}}\boldsymbol{\epsilon}_\theta(\mathbf{z}_t, t)) + \sigma\boldsymbol{\epsilon}$. We sample $\mathbf{z}_T \sim \mathcal{N}(\mathbf{z}_T; \mathbf{0}, \boldsymbol{I})$ from standard Gaussian distribution, by iteratively denoising, finally a data sample $\mathbf{z}_0$ is generated.

## A.2    ALGORITHMS OF TRAINING AND SAMPLING

**Training.**  The training process of the HGLDM consists of two stages, as described in Algorithm 1 in the Appendix A. In the first stage, we train the molecular autoencoder, which includes the molecular encoder $\mathcal{E}_\phi$ and molecular decoder $\mathcal{D}_\psi$. We optimize the objective function $\mathcal{L}_{AE}$, which consists of the reconstruction loss $L_{Rec}$ and the Kullback-Leibler (KL) divergence loss $\mathcal{L}_{KL}$, weighted by a hyperparameter $\gamma$. We continue optimizing until the parameters $\phi$ and $\psi$ have converged.

In the second stage, we train the hierarchical denoising neural network $\boldsymbol{\epsilon}_\theta$ together with the encoder $\mathcal{E}_\phi$. We optimize the objective function $\mathcal{L}$, which is the expected squared error between the denoising neural network's output and the original noise, weighted by the reweighting terms $w(t)$. We continue optimizing until the parameters $\theta$ and $\phi$ have converged.

**Sampling.**  The sampling process of the HGLDM, as described in Algorithm 2 in Appendix A, involves generating new molecules using the trained model. To generate a new molecule, we first sample the initial latent embeddings $\mathbf{z}_T^{\mathbf{X}}, \mathbf{z}_T^{\mathbf{M}}, \mathbf{z}_T^{\mathbf{G}}$ from a standard Gaussian distribution. We then perform a reverse diffusion process by iterating through the diffusion time steps $T$. In each step, we sample a noise vector $\boldsymbol{\epsilon}$ and update the latent embeddings according to the hierarchical denoising neural network $\boldsymbol{\epsilon}_\theta$ and the noise schedule. Finally, we decode the latent embeddings $z_0^{\mathbf{X}}, z_0^{\mathbf{M}}, z_0^{\mathbf{G}}$ using the molecular decoder $\mathcal{D}_\psi$ to obtain the generated molecule's atom features $\mathbf{X}$ and edge features $\mathbf{E}$.

---

**Algorithm 1:** Training

---

**Input:** The molecular graph $G = (\mathbf{X}, \mathbf{E})$, Molecular Encoder $\mathcal{E}_\phi$, Molecular Decoder $\mathcal{D}_\psi$, Hierarchical Denoising Neural Network $\epsilon_\theta$, weight of KL loss $\gamma$, $\bar{\alpha}_t$ related to the noise schedule, reweighting terms $w(t)$.

**First Stage: Molecular Autoencoder Training**

**repeat**

$\quad \boldsymbol{\mu}^\mathbf{X}, \boldsymbol{\mu}^\mathbf{M}, \boldsymbol{\mu}^\mathbf{G} \leftarrow \mathcal{E}_\phi(\mathbf{X}, \mathbf{E})$

$\quad$ Sample $\boldsymbol{\epsilon}^\mathbf{X} \sim \mathcal{N}(\mathbf{0}, \boldsymbol{I}), \boldsymbol{\epsilon}^\mathbf{M} \sim \mathcal{N}(\mathbf{0}, \boldsymbol{I}), \boldsymbol{\epsilon}^\mathbf{G} \sim \mathcal{N}(\mathbf{0}, \boldsymbol{I})$

$\quad \mathbf{z}^\mathbf{X} \leftarrow \boldsymbol{\mu}^\mathbf{X} + \boldsymbol{\sigma}_0 \boldsymbol{\epsilon}^\mathbf{X}, \mathbf{z}^\mathbf{M} \leftarrow \boldsymbol{\mu}^\mathbf{M} + \boldsymbol{\sigma}_0 \boldsymbol{\epsilon}^\mathbf{M}, \mathbf{z}^\mathbf{G} \leftarrow \boldsymbol{\mu}^\mathbf{G} + \boldsymbol{\sigma}_0 \boldsymbol{\epsilon}^\mathbf{G}$

$\quad \hat{\mathbf{X}}, \hat{\mathbf{E}} \leftarrow \mathcal{D}_\psi(\mathbf{z}^\mathbf{X}, \mathbf{z}^\mathbf{M}, \mathbf{z}^\mathbf{G})$

$\quad \mathcal{L}_{AE} = L_{Rec} + \gamma \mathcal{L}_{KL}$

$\quad \phi, \psi \leftarrow optimizer(\mathcal{L}_{AE}, \phi, \psi)$

**until** $\phi$ and $\psi$ converged;

**Second Stage: Hierarchical Denoising Neural Network Training**

**repeat**

$\quad \mathbf{z}_0^\mathbf{X}, \mathbf{z}_0^\mathbf{M}, \mathbf{z}_0^\mathbf{G} \sim q_\phi(\mathbf{z}^\mathbf{X}, \mathbf{z}^\mathbf{M}, \mathbf{z}^\mathbf{G} | \mathbf{X}, \mathbf{E})$

$\quad$ Sample $t \sim \mathbf{U}(0, T), \boldsymbol{\epsilon}^\mathbf{X} \sim \mathcal{N}(\mathbf{0}, \boldsymbol{I}), \boldsymbol{\epsilon}^\mathbf{M} \sim \mathcal{N}(\mathbf{0}, \boldsymbol{I}), \boldsymbol{\epsilon}^\mathbf{G} \sim \mathcal{N}(\mathbf{0}, \boldsymbol{I})$

$\quad \mathbf{z}_t^\mathbf{X} = \sqrt{\bar{\alpha}_t}\mathbf{z}_0^\mathbf{X} + (\sqrt{1 - \bar{\alpha}_t})\boldsymbol{\epsilon}^\mathbf{X}, \mathbf{z}_t^\mathbf{M} = \sqrt{\bar{\alpha}_t}\mathbf{z}_0 + (\sqrt{1 - \bar{\alpha}_t})\boldsymbol{\epsilon}^\mathbf{M}, \mathbf{z}_t^\mathbf{G} = \sqrt{\bar{\alpha}_t}\mathbf{z}_0 + (\sqrt{1 - \bar{\alpha}_t})\boldsymbol{\epsilon}^\mathbf{G}$

$\quad \mathcal{L} = \mathbb{E}_{\mathbf{z}_0^\mathbf{X}, \mathbf{z}_0^\mathbf{M}, \mathbf{z}_0^\mathbf{G}, \boldsymbol{\epsilon} \sim \mathcal{N}(0, I), t}[w(t)||\boldsymbol{\epsilon} - \boldsymbol{\epsilon}_\theta(\mathbf{z}_t^\mathbf{X}, \mathbf{z}_t^\mathbf{M}, \mathbf{z}_t^\mathbf{G}, t)||^2]$, where $\boldsymbol{\epsilon} = [\boldsymbol{\epsilon}^\mathbf{X}, \boldsymbol{\epsilon}^\mathbf{M}, \boldsymbol{\epsilon}^\mathbf{G}]$

$\quad \theta, \phi \leftarrow optimizer(\mathcal{L}; \theta, \phi)$

**until** $\theta, \phi$ converged;

---

**Algorithm 2:** Sampling

---

**Input:** Hierarchical Denoising Neural Network $\epsilon_\theta$, Molecular Decoder $\mathcal{D}_\psi$, diffusion time steps $T$

**Output:** New molecule $G = (\mathbf{X}, \mathbf{E})$

Sample $\mathbf{z}_T^\mathbf{X}, \mathbf{z}_T^\mathbf{M}, \mathbf{z}_T^\mathbf{G} \sim \mathcal{N}(\mathbf{0}, \boldsymbol{I})$

**for** $(t = T, T - 1, \cdots, 1)$ **do**

$\quad$ Sample $\boldsymbol{\epsilon} \sim \mathcal{N}(\mathbf{0}, \mathbf{I})$, where $\boldsymbol{\epsilon} = [\boldsymbol{\epsilon}^\mathbf{X}, \boldsymbol{\epsilon}^\mathbf{M}, \boldsymbol{\epsilon}^\mathbf{G}]$

$\quad \boldsymbol{z}_{t-1} = \frac{1}{\sqrt{1 - \beta_t}}(\mathbf{z}_t - \frac{\beta_t}{\sqrt{1 - \bar{\alpha}_t}}\boldsymbol{\epsilon}_\theta(\mathbf{z}_t, t)) + \sigma\boldsymbol{\epsilon}$, where $\boldsymbol{z}_t = [\boldsymbol{z}_t^\mathbf{X}, \boldsymbol{z}_t^\mathbf{M}, \boldsymbol{z}_t^\mathbf{G}]$

**end**

$\mathbf{X}, \mathbf{E} \leftarrow \mathcal{D}_\psi(\boldsymbol{z}_0^\mathbf{X}, \boldsymbol{z}_0^\mathbf{M}, \boldsymbol{z}_0^\mathbf{G})$

**Return** $\mathbf{X}, \mathbf{E}$

---

## B  IMPLEMENTATION DETAILS

**Datasets.** We utilize four benchmark datasets for our experiments: QM9 (Blum & Reymond, 2009) with 133K molecules (up to 9 heavy atoms), ZINC250K (Irwin et al., 2012) with 250K molecules (up to 38 atoms), GuacaMol (Brown et al., 2019) with 1.3M molecules (up to 88 atoms) and MOSES (Polykovskiy et al., 2020) with more than 1.9M molecules.

**Baselines.** We compare the proposed models (GLDM and HGLDM) with the following state-of-the-art models. **VAE-based models**: PS-VAE (Kong et al., 2022) generates molecules based on mined frequent principal subgraphs from the dataset; **Diffusion-based models**: GDSS (Jo et al., 2022b) proposes a graph diffusion process that models the joint distribution of the nodes and edges through a system of stochastic differential equations (SDEs), DiGress (Vignac et al., 2023) is a discrete denoising diffusion model for generating graphs with categorical node and edge attributes. The implementation details of the proposed methods and baselines are provided in the Appendix B.

**Evaluation Metrics.** We consider six metrics for molecule generation: validity, uniqueness, novelty, diversity, FCD and NSPDK MMD. Validity measures the proportion of molecules that pass basic valency checks. Uniqueness measures the ratio of unique ones in generated molecules. Novelty assesses the ability of the models to generate molecules not contained in the training set. Diversity (Huang et al., 2021) evaluates the internal diversity of a set of molecules. The internal diversity is defined as the average pairwise Tanimoto distance between the Morgan fingerprints. FCD (Fréchet ChemNet Distance) (Preuer et al., 2018) evaluates the distance between the training and generated

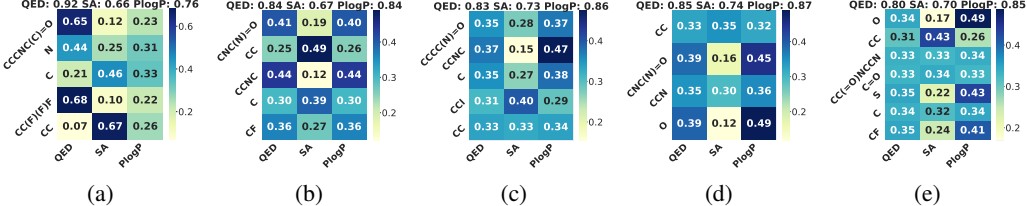

Figure 5: Heatmaps of attention scores between subgraphs and molecular properties, where the horizontal axis represents the names of various molecular properties, and the vertical axis represents the SMILES of subgraphs.

sets using the activations of the penultimate layer of the ChemNet. NSPDK (Neighborhood subgraph pairwise distance kernel) MMD (Costa & De Grave, 2010) is the MMD between the generated molecules and test molecules, which considers both the node and edge features for evaluation.

**Hyperparameters of Hierarchical Autoencoder.** Most of our hyperparameters follow those presented in the Kong et al. (2022) to train the PS-VAE. However, we made a modification for the QM9 dataset by setting the weight parameter for the KL loss to 1e-5 to achieve better performance. This change may be attributed to the limited variety of atom types in the QM9 dataset; a lower KL loss can contribute to increased uniqueness and novelty in the generated samples.

**Hyperparameters of Latent Diffusion Model for Hierarchical Graph.** We set the diffusion steps to 200 and stacked 6 hierarchical blocks in the denoising model. The hidden layer dimensions of the hierarchical embeddings are set to 64, and we use 8 attention heads to compute the attention scores. The learning rate and batch size are set to 1e-4 and 128, respectively.

### B.1 UNCONDITIONAL GENERATION

The training processes of all the methods are carried out using Distributed Data Parallel (DDP) on 8 NVIDIA V100 GPUs for a total of 100 epochs, while the sampling processes are performed on a single NVIDIA V100 GPU. We generate a total of 10,000 molecules for each method during the sampling phase.

### B.2 CONDITIONAL GENERATION

**Properties and Oracles.** (1) QED (Quantitative Estimate of Drug-likeness) (Bickerton et al., 2012) measures the drug-likeness of molecules with a range from 0 to 1; (2) SA (Synthetic Accessibility) (Ertl & Schuffenhauer, 2009) is a metric that evaluates how easily a molecule can be synthesized; (3) PlogP (Penalized logP) (Kusner et al., 2017) is logP penalized by synthesis accessibility and ring size which has an unbounded range.

## C ADDITIONAL EXPERIMENTAL RESULTS

### C.1 ABLATION STUDY ON CONDITIONAL GENERATION

**Relationship between subgraphs and molecular properties.** By incorporating hierarchical embeddings, HGLDM demonstrates its ability to explore the relationship between subgraphs and molecular properties. Examining the attention scores between subgraphs and molecular properties of the 5 molecules presented in Figure 5 as representative cases, we make the following observations: Firstly, there is a strong correlation between molecular properties and the presence of influential subgraphs. For example, in Figure 5a, the molecule has a higher QED value due to the presence of subgraphs with higher QED influences. Similarly, the molecule in Figure 5b has a lower SA value, while the molecules in Figures 5c-5e exhibit higher PlogP values due to the influential subgraphs. Secondly, certain specific subgraphs show a strong correlation with molecular properties. For instance, longer subgraphs consistently exhibit more significant QED influence, while 'CC' subgraphs have a more substantial impact on the SA value. Moreover, 'CCNC', 'CNC(N)=O', and 'O' subgraphs (atoms) are more influential on PlogP value. These observations highlight the impor-

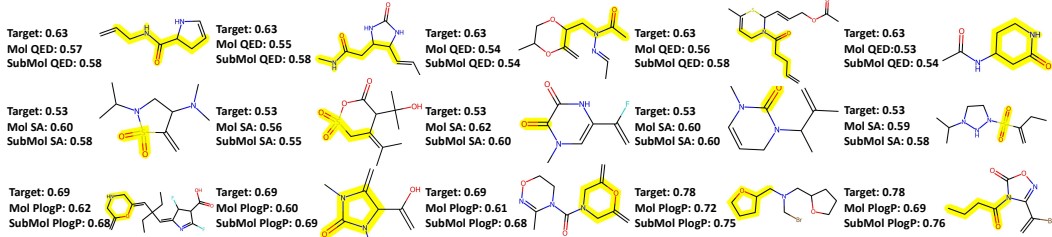

Figure 6: Molecule samples of conditional generation on ZINC250K.

tance of considering atom-level and subgraph-level information in understanding the relationships between molecular properties and molecular structures.

**Generated Molecule Examples.** In Figure 6, we present the molecules generated through conditional generation based on QED, SA, and PlogP conditions. By comparing the target values with the QED, SA, and PlogP values of both the molecules (Mol) and the subgraph of the molecules (SubMol), we observe that the SubMol values are closer to the target values. This highlights how the hierarchical structure can identify subgraphs that match the target values, leading to the generation of molecules with properties that are more aligned with the desired targets.

## D RELATED WORK

**Diffusion-based Molecule Generation.** In recent years, generative models based on diffusion processes have emerged as a promising direction for the automated design of molecules (Liu et al., 2023; Lee et al., 2023). Most of the previous diffusion models have the mismatch problem of continuous diffusion space and discrete data space (Niu et al., 2020). To address this limitation, some recent works have proposed to use discrete noise instead of Gaussian noise for graph diffusion models. Among them, Haefeli et al. (2022) designed a discrete denoising diffusion kernel for unattributed graphs and largely accelerated the sampling process. Chen et al. (2022) proposed the NVDiff, which only diffuses the node variables and decodes edge types from them. Similarly, Di-Gress (Vignac et al., 2023) is a discrete diffusion model that progressively edits a graph by adding or removing edges, and changing the categories. Another recent work (Luo et al., 2022) proposed a fast graph generation model via spectral diffusion, which leverages low-rank diffusion SDEs on the graph spectrum space instead of the whole graph adjacency matrix space. Besides those methods that operate on graph structures, there are also several works that define diffusion models for molecule generation in 3D atomic positions (Trippe et al., 2022; Hoogeboom et al., 2022; Bao et al., 2022; Huang et al., 2022; Qiang et al., 2023; Xu et al., 2023). To the best of our knowledge, our proposed method is the first work that incorporates the hierarchical structure information into the design of discrete denoising diffusion model for the molecule generation.

**Hierarchical Graph Learning.** Hierarchical graph learning extends GNNs by incorporating multiple levels of abstraction and resolution in the graph, which can capture both local and global information about the graph (Gao et al., 2023). To construct multiple levels of graph hierarchy, Defferrard et al. (2016) employed graph coarsening algorithms. Ying et al. (2018); Gao & Ji (2019) proposed to jointly learn the graph hierarchy and the encoding process. It was only recently that the power of graph hierarchy in graph generation has been explored (Jin et al., 2020; Kuznetsov & Polykovskiy, 2021; Qu & Zou, 2022; Karami & Luo, 2023). Despite some differences, all of these methods learn the hierarchy for graph generation using VAEs, GANs, or normalizing flows. In contrast, our hierarchy is constructed to incorporate the diffusion model on graphs in the discrete space.

## E LIMITATIONS

Although this paper proposes a hierarchical latent diffusion model that significantly improves upon VAE-based models, there are still some limitations. Firstly, the model's performance is still subject to the limitations of the VAE backbone, so choosing a good VAE backbone is crucial for HGLDM. Secondly, while the model greatly reduces training and sampling time compared to other diffusion-based models, it still requires a long training time on large-scale datasets such as MOSES and GuacaMol, which limits the process of hyperparameter tuning and selection.

