# OpenReview forum: "Hierarchical Graph Latent Diffusion Model for Molecule Generation"
_ICLR.cc/2024/Conference — Submitted to ICLR 2024_

### Official Review · Reviewer_y9cz · 2023-10-30

**Soundness:** 2 fair
**Presentation:** 3 good
**Contribution:** 3 good
**Rating:** 5
**Confidence:** 3

**Summary:**

This work presents a novel hierarchical latent diffusion model for molecular graph generation. To be specific, this work introduces GLDM, a latent diffusion model for graphs using graph-level embeddings, and proposes HGLDM, a latent diffusion model that further incorporates structural information, for which these approaches enable efficient training and sampling while outperforming previous graph diffusion models.

**Strengths:**

- The paper is well-written and easy to follow.

- The motivation for using the latent approach, i.e., overcoming the mismatch between the continuous diffusion space and discrete data space and further reducing computational cost, is clear.

- Using hierarchical embeddings of graphs for graph latent diffusion is novel and shows improvements in conditional molecule generation tasks compared to the naive latent diffusion model (GLDM) as well as previous diffusion models.

**Weaknesses:**

- Although this work states that the (hierarchical) latent approach for graph generation provides a scalable solution for molecule generation, the provided experiments are limited to datasets (e.g., GuacaMol) in which previous diffusion models (e.g., GDSS and DiGress) are applicable. In order to justify the scalability of the proposed method, it should be evaluated in a larger dataset.

- The experimental setting for evaluating the computational efficiency is not clear. Is the training and sampling time measured in the same condition, e.g., training conducted via DDP and using the same number of V100 GPUs?

- Generation performance on unconditional molecule generation tasks should be evaluated with more descriptive metrics, for example, FCD, Scaffold similarity [1], and Fragment similarity [1]. Reported metrics, i.e., validity, uniqueness, and novelty fail to measure how similar (e.g., chemical aspects) are the generated molecules to the molecules from the test set. In particular, under the current setting, GDSS seems to be showing comparable results in large datasets (ZINC250K and GuacaMol) with significantly fewer parameters.

- The quantitative results of Tables 2 and 3 show that the performances of GLDM and HGLDM on unconditional generation tasks are almost the same, whereas there is a significant improvement using the hierarchical approach for conditional generation tasks. What is the reason for the hierarchical approach only effective in conditional tasks?

- As the continuous diffusion model (e.g., GDSS) outperforms the discrete diffusion model (e.g., DiGress) in Table 2, the continuous diffusion model should be compared as a baseline in Table 3 (i.e., conditional generation task). Although GDSS does not explicitly present a conditional framework, recent work [2] proposes a conditional molecule generation framework using classifier guidance based on GDSS, which could be used as a baseline.

- The performance of GLDM (and HGLDM) comes from the effectiveness of using a latent representation of graphs compared to previous graph diffusion models, not from the diffusion processes. Thereby, analysis of the latent representation, e.g., interpolation in the latent space or clustering of the latent points with respect to certain conditions, would greatly strengthen this work.

- Missing references on related works:
  - Qiang et al., Coarse-to-Fine: a Hierarchical Diffusion Model for Molecule Generation in 3D, ICML 2023
  - Xu et al., Geometric Latent Diffusion Models for 3D Molecule Generation, ICML 2023

- I would like to raise my score if the above concerns are sufficiently addressed.

---

[1] Polykovskiy et al., Molecular Sets (MOSES): A Benchmarking Platform for Molecular Generation Models, arXiv 2018
[2] Lee et al., Exploring Chemical Space with Score-based Out-of-distribution Generation, ICML 2023

**Questions:**

- Please address the questions in the Weakness.

- Is the results of Table 2 from a single run or an average of multiple runs?

---

> ### Author Response · Authors · 2023-11-21
> **Response to Reviewer y9cz**
>
> We thank the reviewer for the constructive and detailed comments. The reviewer has provided a nice suggestion by including the analysis of the latent representation. We agree that it will better emphasize the contribution of our work, and have included this experiment in the revised version. The response for each concern has been provided as follows.
>
> ---
> ### Weaknesses:
> **W1. Although this work states that the (hierarchical) latent approach for graph generation provides a scalable solution for molecule generation, the provided experiments are limited to datasets (e.g., GuacaMol) in which previous diffusion models (e.g., GDSS and DiGress) are applicable. In order to justify the scalability of the proposed method, it should be evaluated in a larger dataset.**
>
> R1: Thank you for your suggestion. During the rebuttal period, we have added experimental results on a larger molecule dataset, MOSES dataset. The results are shown in the Table below. Our proposed method achieved better Valid, Novelty, Diversity and comparable Unique scores compared to the baseline methods on the MOSES dataset.
>
> **Table 1: Unconditional Generation Results on MOSES Dataset**
> | Methods | Valid $\uparrow$           | Unique$\uparrow$           | Novelty$\uparrow$          | Diversity$\uparrow$        | FCD$\downarrow$             | NSPDK MMD$\downarrow$      |
> |-----------------|------------------------------------|------------------------------------|------------------------------------|------------------------------------|-------------------------------------|------------------------------------|
>  | GDSS    | **1.000 $\pm$ 0.000** | 0.994 $\pm$ 0.003          | 0.999 $\pm$ 0.000          | 0.899 $\pm$ 0.000          | 21.265 $\pm$ 0.249          | 0.037 $\pm$ 0.005          |
> | DiGress | 0.858 $\pm$ 0.005          | **1.000 $\pm$ 0.000** | 0.996 $\pm$ 0.001          | 0.886 $\pm$ 0.000          | **9.228 $\pm$ 0.081**  | **0.010 $\pm$ 0.000** |
>  | PS-VAE  | **1.000 $\pm$ 0.000** | 0.999 $\pm$ 0.000          | **1.000 $\pm$ 0.000** | 0.905 $\pm$ 0.000          | 26.401 $\pm$ 0.078          | 0.079 $\pm$ 0.000          |
>  | GLDM    | **1.000 $\pm$ 0.000** | 0.998 $\pm$ 0.000          | **1.000 $\pm$ 0.000** | 0.905 $\pm$ 0.000          | 26.365 $\pm$ 0.095          | 0.077 $\pm$ 0.001          |
>  | HGLDM   | **1.000 $\pm$ 0.000** | 0.999 $\pm$ 0.000          | **1.000 $\pm$ 0.000** | **0.906 $\pm$ 0.000** | 25.815 $\pm$ 0.053          | 0.072 $\pm$ 0.000          |
>
> **W2. The experimental setting for evaluating the computational efficiency is not clear. Is the training and sampling time measured in the same condition, e.g., training conducted via DDP and using the same number of V100 GPUs?**
>
> R2: Sorry for the confusion. The training and sampling time are measured under the same conditions. We have clarified this in the revised version.
>
> **W3. Generation performance on unconditional molecule generation tasks should be evaluated with more descriptive metrics, for example, FCD, Scaffold similarity [1], and Fragment similarity [1]. Reported metrics, i.e., validity, uniqueness, and novelty fail to measure how similar (e.g., chemical aspects) are the generated molecules to the molecules from the test set. In particular, under the current setting, GDSS seems to be showing comparable results in large datasets (ZINC250K and GuacaMol) with significantly fewer parameters.**
>
> R3: Thank you for your suggestion. In the revised version, we have added FCD and NSPDK to evaluate the graph distribution of the generated molecules. The detailed results are shown in the Tables 1-4 below. Although the proposed method does not achieve best results in FCD and NSPDK, this is mainly limited by the VAE backbone model, and we have improved upon PSVAE in terms of  FCD and NSPDK across all the datasets. Furthermore, unlike general graph generation that primarily focuses on learning graph distributions, we believe that comparing distribution similarity in molecular generation tasks without considering molecular properties lacks practical significance. Therefore, after verifying that the performance of the proposed method in terms of metrics such as validity and novelty are comparable, indicating that our model performs well in unconditional generation, we shift our focus to conditional molecular generation  that can incorporate more conditional information into the generation process.

---

> > ### Author Response · Authors · 2023-11-21
> >
> > **Table 1: Unconditional Generation Results on MOSES Dataset**
> > | Methods | Valid $\uparrow$           | Unique$\uparrow$           | Novelty$\uparrow$          | Diversity$\uparrow$        | FCD$\downarrow$             | NSPDK MMD$\downarrow$      |
> > |-----------------|------------------------------------|------------------------------------|------------------------------------|------------------------------------|-------------------------------------|------------------------------------|
> >  | GDSS    | **1.000 $\pm$ 0.000** | 0.994 $\pm$ 0.003          | 0.999 $\pm$ 0.000          | 0.899 $\pm$ 0.000          | 21.265 $\pm$ 0.249          | 0.037 $\pm$ 0.005          |
> > | DiGress | 0.858 $\pm$ 0.005          | **1.000 $\pm$ 0.000** | 0.996 $\pm$ 0.001          | 0.886 $\pm$ 0.000          | **9.228 $\pm$ 0.081**  | **0.010 $\pm$ 0.000** |
> >  | PS-VAE  | **1.000 $\pm$ 0.000** | 0.999 $\pm$ 0.000          | **1.000 $\pm$ 0.000** | 0.905 $\pm$ 0.000          | 26.401 $\pm$ 0.078          | 0.079 $\pm$ 0.000          |
> >  | GLDM    | **1.000 $\pm$ 0.000** | 0.998 $\pm$ 0.000          | **1.000 $\pm$ 0.000** | 0.905 $\pm$ 0.000          | 26.365 $\pm$ 0.095          | 0.077 $\pm$ 0.001          |
> >  | HGLDM   | **1.000 $\pm$ 0.000** | 0.999 $\pm$ 0.000          | **1.000 $\pm$ 0.000** | **0.906 $\pm$ 0.000** | 25.815 $\pm$ 0.053          | 0.072 $\pm$ 0.000          |
> >
> > **Table 2: Unconditional Generation Results on QM9 Dataset**
> >  Methods | Valid $\uparrow$           | Unique$\uparrow$           | Novelty$\uparrow$          | Diversity$\uparrow$        | FCD$\downarrow$             | NSPDK MMD$\downarrow$      |
> > |-----------------|------------------------------------|------------------------------------|------------------------------------|------------------------------------|-------------------------------------|------------------------------------|
> > | GDSS    | 0.957 $\pm$ 0.000          | 0.982 $\pm$ 0.003          | 0.988 $\pm$ 0.001          | **0.925 $\pm$ 0.000** | 2.959 $\pm$ 0.040           | 0.003 $\pm$ 0.000          |
> > | DiGress | 0.992 $\pm$ 0.003          | 0.960 $\pm$ 0.001          | 0.391 $\pm$ 0.001          | 0.920 $\pm$ 0.000          | **2.123 $\pm$ 0.033**  | **0.001 $\pm$ 0.000** |
> > | PS-VAE  | **1.000 $\pm$ 0.000** | 0.981 $\pm$ 0.002          | 0.996 $\pm$ 0.000          | 0.881 $\pm$ 0.000          | 16.877 $\pm$ 0.059          | 0.059 $\pm$ 0.000          |
> > | GLDM    | **1.000 $\pm$ 0.000** | 0.982 $\pm$ 0.001          | 0.996 $\pm$ 0.000          | 0.884 $\pm$ 0.000          | 14.829 $\pm$ 0.169          | 0.050 $\pm$ 0.001          |
> > | HGLDM   | **1.000 $\pm$ 0.000** | **0.985 $\pm$ 0.002** | **0.997 $\pm$ 0.000** | 0.884 $\pm$ 0.000          | 14.576 $\pm$ 0.183          | 0.047 $\pm$ 0.001          |
> >
> > **Table 3: Unconditional Generation Results on ZINC250K Dataset**
> > | Methods | Valid $\uparrow$           | Unique$\uparrow$           | Novelty$\uparrow$          | Diversity$\uparrow$        | FCD$\downarrow$             | NSPDK MMD$\downarrow$      |
> > |-----------------|------------------------------------|------------------------------------|------------------------------------|------------------------------------|-------------------------------------|------------------------------------|
> > | GDSS    | **1.000 $\pm$ 0.000** | 0.997 $\pm$ 0.001          | **1.000 $\pm$ 0.000** | 0.902 $\pm$ 0.000          | 16.086 $\pm$ 0.071          | **0.018 $\pm$ 0.000** |
> > | DiGress | 0.565 $\pm$ 0.005          | **1.000 $\pm$ 0.000** | **1.000 $\pm$ 0.000** | 0.882 $\pm$ 0.000          | **13.042 $\pm$ 0.164** | 0.031 $\pm$ 0.001          |
> >  | PS-VAE  | **1.000 $\pm$ 0.000** | 0.993 $\pm$ 0.001          | **1.000 $\pm$ 0.000** | 0.912 $\pm$ 0.000          | 20.386 $\pm$ 0.061          | 0.085 $\pm$ 0.001          |
> > | GLDM    | **1.000 $\pm$ 0.000** | 0.994 $\pm$ 0.001          | **1.000 $\pm$ 0.000** | 0.913 $\pm$ 0.000          | 20.444 $\pm$ 0.088          | 0.086 $\pm$ 0.001          |
> >  | HGLDM   | **1.000 $\pm$ 0.000** | 0.997 $\pm$ 0.001          | **1.000 $\pm$ 0.000** | **0.914 $\pm$ 0.000** | 19.913 $\pm$ 0.091          | 0.084 $\pm$ 0.000          |

---

> > > ### Author Response · Authors · 2023-11-21
> > >
> > > **Table 4: Unconditional Generation Results on Guacamol Dataset**
> > >  | Methods | Valid $\uparrow$           | Unique$\uparrow$           | Novelty$\uparrow$          | Diversity$\uparrow$        | FCD$\downarrow$             | NSPDK MMD$\downarrow$      |
> > > |-----------------|------------------------------------|------------------------------------|------------------------------------|------------------------------------|-------------------------------------|------------------------------------|
> > > | GDSS    | **1.000 $\pm$ 0.000** | 0.986 $\pm$ 0.001          | 0.996 $\pm$ 0.001          | 0.892 $\pm$ 0.000          | 40.291 $\pm$ 0.072          | 0.058 $\pm$ 0.000          |
> > > | DiGress | 0.875 $\pm$ 0.005          | **1.000 $\pm$ 0.000** | **0.999 $\pm$ 0.001** | 0.904 $\pm$ 0.000          | **12.069 $\pm$ 0.051** | **0.018 $\pm$ 0.000** |
> > >  | PS-VAE  | **1.000 $\pm$ 0.000** | 0.998 $\pm$ 0.000          | 0.998 $\pm$ 0.000          | **0.905 $\pm$ 0.000** | 24.105 $\pm$ 0.082          | 0.090 $\pm$ 0.000          |
> > >  | GLDM    | **1.000 $\pm$ 0.000** | 0.998 $\pm$ 0.000          | 0.998 $\pm$ 0.000          | 0.904 $\pm$ 0.000          | 23.879 $\pm$ 0.041          | 0.095 $\pm$ 0.000          |
> > > | HGLDM   | **1.000 $\pm$ 0.000** | 0.999 $\pm$ 0.001          | **0.999 $\pm$ 0.000** | **0.905 $\pm$ 0.000** | 23.845 $\pm$ 0.098          | 0.095 $\pm$ 0.001          |
> > >
> > >
> > > **W4. The quantitative results of Tables 2 and 3 show that the performances of GLDM and HGLDM on unconditional generation tasks are almost the same, whereas there is a significant improvement using the hierarchical approach for conditional generation tasks. What is the reason for the hierarchical approach only effective in conditional tasks?**
> > >
> > > R4: Because the properties of molecules are often determined by distinctive functional groups (subgraphs), HGLDM performs better than GLDM in conditional generation tasks by capturing subgraph information. We have inclued this explanation in the revised version.
> > >
> > > **W5. As the continuous diffusion model (e.g., GDSS) outperforms the discrete diffusion model (e.g., DiGress) in Table 2, the continuous diffusion model should be compared as a baseline in Table 3 (i.e., conditional generation task). Although GDSS does not explicitly present a conditional framework, recent work [2] proposes a conditional molecule generation framework using classifier guidance based on GDSS, which could be used as a baseline.**
> > >
> > > R5: Thank you for your suggestion. We have incorporated the conditional generation results of GDSS in the revised version. The updated results are shown in Table 5 below. Through comparison, we found that our proposed model outperforms GDSS in single property conditional generation. However, in multi-properties  generation, although GDSS performs the best, its conditional generation results are slightly worse compared to unconditional generation. Additionally, GDSS exhibits significant variance in unconditional generation, resulting in unstable generated results.
> > >
> > > **Table 5: MAE results of conditional generation on ZINC250K.**
> > > | Models                           | Task                  | QED                   | SA                    | PlogP                 | QED\&SA\&PlogP |
> > > |----------------------------------|-----------------------|-----------------------|-----------------------|-----------------------|-------------------------------|
> > > | GDSS | Unconditional | 0.437(0.006)  | 0.058(0.025)  | 0.114(0.044)  | **0.176(0.027)** |
> > > | GDSS | Conditional  | 0.435(0.009)  | 0.033(0.007)  | 0.077(0.020)  | 0.182(0.012)          |
> > > | DiGress      | Unconditional         | 0.859(0.009)          | 0.679(0.010)          | 0.788(0.013)          | 0.775(0.010)                  |
> > > |     DiGress       | Conditional           | 0.531(0.018)          | 0.152(0.011)          | 0.173(0.024)          | 0.207(0.019)                  |
> > > | GLDM       | Unconditional         | 0.414(0.005)          | 0.065(0.008)          | 0.099(0.012)          | 0.191((0.008)                 |
> > > |  GLDM           | Conditional           | 0.404(0.005)          | 0.063(0.010)          | 0.094(0.011)          | 0.187(0.009)                  |
> > > | HGLDM        | Unconditional         | 0.415(0.005)          | 0.063(0.013)          | 0.085(0.010)          | 0.188(0.009)                  |
> > > |    HGLDM      | Conditional           | **0.348(0.018)** | **0.021(0.010)** | **0.068(0.004)** | 0.185(0.008)                  |

---

> > > > ### Author Response · Authors · 2023-11-21
> > > >
> > > > **W6. The performance of GLDM (and HGLDM) comes from the effectiveness of using a latent representation of graphs compared to previous graph diffusion models, not from the diffusion processes. Thereby, analysis of the latent representation, e.g., interpolation in the latent space or clustering of the latent points with respect to certain conditions, would greatly strengthen this work.**
> > > >
> > > > R6: Thank you for your suggestion. During the rebuttal period, we have conducted a new experiment on the interpolation of latent embeddings. we randomly sample two latent embeddings, $\mathbf{z}^{G_1}$ and $\mathbf{z}^{G_2}$, and generate the interpolated embeddings $G'$ through $\mathbf{z}^{G'}=\lambda\mathbf{z}^{G_1}+(1-\lambda)\mathbf{z}^{G_2}$, where $\lambda \in (0, 1)$. Based on the visualized results of these latent embeddings (Figure 4 in the revised version), we can draw two key conclusions. Firstly, the clear boundaries between embeddings of different properties in the scatter plot, along with the high property values retained by the interpolated embeddings, suggest that **latent embeddings can effectively distinguish between molecules with varying property values**. This provides a solid foundation for further conditional generation using these latent embeddings. Secondly, **latent embeddings trained by HGLDM capture more distinct features of different property values compared to PS-VAE latent embeddings**, as evidenced by the farther distances between HGLDM latent embeddings with different property values and the higher property values of molecules decoded from HGLDM latent embeddings interpolation. This highlights the enhancement of the latent diffusion framework over VAE-based models (Please refer to General Response G1 and Section 5.2 in the revised version  for more details).
> > > >
> > > > **W7. Missing references on related works:
> > > > -Qiang et al., Coarse-to-Fine: a Hierarchical Diffusion Model for Molecule Generation in 3D, ICML 2023
> > > > -Xu et al., Geometric Latent Diffusion Models for 3D Molecule Generation, ICML 2023**
> > > >
> > > > R7: Thank you for your suggestion. We have added these references to the Related Work section in the revised version.
> > > >
> > > > ---
> > > > ### Questions:
> > > > **Q1. Is the results of Table 2 from a single run or an average of multiple runs?**
> > > >
> > > > R1: The original results of Table 2 is from a single run. During the rebuttal period, we have conducted eight samplings with  different seeds and reported the mean and variance in the revised version. The new results are shown in the Tables 1-4 above.

---

> > > > > ### Comment · Reviewer_y9cz · 2023-11-23
> > > > > **Thank you for the response**
> > > > >
> > > > > Thank you for the detailed response and additional experiments.
> > > > >
> > > > > - The results of MOSES do not show improvements over the baselines: Validity, Uniqueness, and Novelty are similar to GDSS with a small improvement in Diversity (0.7 pt) while a significant increase in FCD and NSPDK.
> > > > >
> > > > > - The FCD and NSPDK results on all the datasets are significantly worse than the baselines, indicating that the proposed method is not capable of learning the data distribution. Validity, Uniqueness, and Novelty are similar to that of GDSS with no clear improvement.
> > > > >
> > > > > - In Table 3, GDSS performs the best on multi-properties which is the most practical setting for real-world tasks, for example, drug discovery where it is required to satisfy many different objectives. In my opinion, performing better in a single property generation does not provide much merit.
> > > > >
> > > > > Due to these concerns, I maintain my score.

---

### Official Review · Reviewer_vEbk · 2023-10-30

**Soundness:** 3 good
**Presentation:** 2 fair
**Contribution:** 1 poor
**Rating:** 3
**Confidence:** 4

**Summary:**

This paper proposes a latent diffusion model that aims to generate hierarchical levels of latent variables such as node-level, subgraph-level, and graph-level, simultaneously. To construct the latent space, the authors leverage the PS-VAE where the decoder converts the graph-level latent variables to the molecule by sequentially predicting the fragments and then predicting the links between fragments. To generate the latent variables of node-, subgraph-, and graph-level, the authors leverage the generative process of DDPM. The authors propose an architecture that models the dependency between node-, subgraph-, and graph-level embeddings alleviating the burden of considering the edge features. The proposed method is evaluated on the molecule generation tasks in the conditional and unconditional settings.

**Strengths:**

* The auxiliary generation of node and subgraph embeddings can enrich the forwarded information of the diffusion process. Specifically, even though PS-VAE requires only the graph embedding in the decoding stage, the authors propose to generate the node and subgraph embedding along with the graph embedding. Defining the correlated generative processes in this paper could convey more information to the model.

**Weaknesses:**

* It is not clear why the authors select a way to generate graph embeddings and then decode them. Accessing the graph embedding could contain less information than accessing the subgraph- or node- and edge-level latent variables. The
* The name of the proposed method is misleading. The goal of the proposed method is to generate the hierarchical latent variables. However, the name (Hierarchical Graph Latent Diffusion Model) can be misinterpreted as a sequence of the diffusion models to generate the latent variables.
* The authors report the validity, uniqueness, and novelty as the main results. However, these metrics seem to be restricted to only measure the sample diversity.
>- For the validity, it is unfair to compare with the denoising diffusion model such as GDSS and DiGress, as the decoding stage of the proposed method and PS-VAE intrinsically do the validity check while linking the fragments. Therefore, reporting and comparing the validity are not enough to demonstrate the effectiveness of the proposed method.
>- For the uniqueness and novelty, they demonstrate that the generative model can guarantee sample diversity. However, to demonstrate whether the generative model precisely learns the data distribution, reporting the uniqueness and the novelty is not enough. Please note that recent works [1,2] leverage FCD, Scaffold similarity, SNN and NSPDK to measure the difference of the generated distribution and the data distribution. Therefore, I believe that measuring the uniqueness and novelty is important, but to demonstrate the effectiveness, it would be better to measure the distributions of the generated molecules.
* The efficiency of the proposed methods seems to come from the light model architecture with smaller dimensions than the model architecture used in the DiGress.

[1] Vignac, Clement, et al. "Digress: Discrete denoising diffusion for graph generation." arXiv preprint arXiv:2209.14734 (2022).

[2] Jo, Jaehyeong, Seul Lee, and Sung Ju Hwang. "Score-based generative modeling of graphs via the system of stochastic differential equations." International Conference on Machine Learning. PMLR, 2022.

**Questions:**

For clarification, I would appreciate if the authors provided an explanation of my questions.
1. In Section 4.1.1, to my understanding, the number of subgraph embeddings should not be the number of nodes. If so, how do you sample the number of subgraph embeddings at the beginning of the sampling stage?
2. How do you get the subgraph embeddings from the PS-VAE architecture?
3. In Table 3, how did you measure the mean absolute error (MAE) on the unconditional setting? Does it mean training on the selected 100 molecules without the conditions?
4. Why are some reported values different from the original papers? For example, the novelty and uniqueness of PS-VAE on the QM9 dataset and the validity of GDSS on the ZINC250k dataset.

---

> ### Author Response · Authors · 2023-11-21
> **Response to Reviewer vEbk**
>
> We thank the reviewer for the provided comments. Regarding Weakness 1, we hope that we have clarified the model design and main contributions of our work to address the misunderstandings. With regards to Weaknesses 2 and 3, we have made revisions in accordance with the reviewer's requests in the revised version. We sincerely hope that the reviewer will raise any further questions if our responses are still unclear. The response for each concern has been provided as follows.
>
> ---
> ### Weaknesses:
> **W1. It is not clear why the authors select a way to generate graph embeddings and then decode them. Accessing the graph embedding could contain less information than accessing the subgraph- or node- and edge-level latent variables.**
>
> R1: We agree with your viewpoint that "Accessing the graph embedding could contain less information than accessing the subgraph- or node- and edge-level latent variables."  We need to clarify that **the main contribution of this paper is the HGLDM model, which includes graph embeddings, subgraph embeddings, and node embeddings.** The reason why edge-level embeddings are not included is that, as demonstrated by the baseline methods in the Section 5.1, the diffusion process on the edge matrix is computationally expensive. The graph embedding-based model GLDM proposed in this paper is just a straightforward attempt inspired by the latent diffusion model, as there exists the decoder that can generate molecules from graph embeddings. However, in this paper, we found that the hierarchical embeddings proposed in this paper are more effective than the graph embeddings, as the latter contain less information, as you mentioned.
>
> **W2. The name of the proposed method is misleading. The goal of the proposed method is to generate the hierarchical latent variables. However, the name (Hierarchical Graph Latent Diffusion Model) can be misinterpreted as a sequence of the diffusion models to generate the latent variables.**
>
> R2: Thank you for your suggestion. We have modified it to "Latent Diffusion Model for Hierarchical Graph" in the revised version to avoid potential misleading.
>
> **W3. The authors report the validity, uniqueness, and novelty as the main results. However, these metrics seem to be restricted to only measure the sample diversity.
> For the validity, it is unfair to compare with the denoising diffusion model such as GDSS and DiGress, as the decoding stage of the proposed method and PS-VAE intrinsically do the validity check while linking the fragments. Therefore, reporting and comparing the validity are not enough to demonstrate the effectiveness of the proposed method.
> For the uniqueness and novelty, they demonstrate that the generative model can guarantee sample diversity. However, to demonstrate whether the generative model precisely learns the data distribution, reporting the uniqueness and the novelty is not enough. Please note that recent works [1,2] leverage FCD, Scaffold similarity, SNN and NSPDK to measure the difference of the generated distribution and the data distribution. Therefore, I believe that measuring the uniqueness and novelty is important, but to demonstrate the effectiveness, it would be better to measure the distributions of the generated molecules.**
>
>
> R3: For the validity, we need to clarify that during the decoding stage, the PS-VAE only performs link prediction to connect fragments and does not perform validity check. In the revised version, we have added FCD and NSPDK to evaluate the graph distribution of the generated molecules. The new results are shown in Tables 1-4 below. Although the proposed method does not achieve best results in FCD and NSPDK, this is mainly limited by the VAE backbone model, and we have improved upon PSVAE in terms of  FCD and NSPDK across all the datasets. **Furthermore, unlike general graph generation that primarily focuses on learning graph distributions, we believe that comparing distribution similarity in molecular generation tasks without considering molecular properties lacks practical significance**. Therefore, after verifying that the performance of the prpoposed method in terms of metrics such as validity and novelty are comparable, indicating that our model performs well in unconditional generation, we shift our focus to conditional molecular generation  that can incorporate more conditional information into the generation process.

---

> > ### Author Response · Authors · 2023-11-21
> >
> > **Table 1: Unconditional Generation Results on QM9 Dataset**
> >  Methods | Valid $\uparrow$           | Unique$\uparrow$           | Novelty$\uparrow$          | Diversity$\uparrow$        | FCD$\downarrow$             | NSPDK MMD$\downarrow$      |
> > |-----------------|------------------------------------|------------------------------------|------------------------------------|------------------------------------|-------------------------------------|------------------------------------|
> > | GDSS    | 0.957 $\pm$ 0.000          | 0.982 $\pm$ 0.003          | 0.988 $\pm$ 0.001          | **0.925 $\pm$ 0.000** | 2.959 $\pm$ 0.040           | 0.003 $\pm$ 0.000          |
> > | DiGress | 0.992 $\pm$ 0.003          | 0.960 $\pm$ 0.001          | 0.391 $\pm$ 0.001          | 0.920 $\pm$ 0.000          | **2.123 $\pm$ 0.033**  | **0.001 $\pm$ 0.000** |
> > | PS-VAE  | **1.000 $\pm$ 0.000** | 0.981 $\pm$ 0.002          | 0.996 $\pm$ 0.000          | 0.881 $\pm$ 0.000          | 16.877 $\pm$ 0.059          | 0.059 $\pm$ 0.000          |
> > | GLDM    | **1.000 $\pm$ 0.000** | 0.982 $\pm$ 0.001          | 0.996 $\pm$ 0.000          | 0.884 $\pm$ 0.000          | 14.829 $\pm$ 0.169          | 0.050 $\pm$ 0.001          |
> > | HGLDM   | **1.000 $\pm$ 0.000** | **0.985 $\pm$ 0.002** | **0.997 $\pm$ 0.000** | 0.884 $\pm$ 0.000          | 14.576 $\pm$ 0.183          | 0.047 $\pm$ 0.001          |
> >
> > **Table 2: Unconditional Generation Results on ZINC250K Dataset**
> > | Methods | Valid $\uparrow$           | Unique$\uparrow$           | Novelty$\uparrow$          | Diversity$\uparrow$        | FCD$\downarrow$             | NSPDK MMD$\downarrow$      |
> > |-----------------|------------------------------------|------------------------------------|------------------------------------|------------------------------------|-------------------------------------|------------------------------------|
> > | GDSS    | **1.000 $\pm$ 0.000** | 0.997 $\pm$ 0.001          | **1.000 $\pm$ 0.000** | 0.902 $\pm$ 0.000          | 16.086 $\pm$ 0.071          | **0.018 $\pm$ 0.000** |
> > | DiGress | 0.565 $\pm$ 0.005          | **1.000 $\pm$ 0.000** | **1.000 $\pm$ 0.000** | 0.882 $\pm$ 0.000          | **13.042 $\pm$ 0.164** | 0.031 $\pm$ 0.001          |
> >  | PS-VAE  | **1.000 $\pm$ 0.000** | 0.993 $\pm$ 0.001          | **1.000 $\pm$ 0.000** | 0.912 $\pm$ 0.000          | 20.386 $\pm$ 0.061          | 0.085 $\pm$ 0.001          |
> > | GLDM    | **1.000 $\pm$ 0.000** | 0.994 $\pm$ 0.001          | **1.000 $\pm$ 0.000** | 0.913 $\pm$ 0.000          | 20.444 $\pm$ 0.088          | 0.086 $\pm$ 0.001          |
> >  | HGLDM   | **1.000 $\pm$ 0.000** | 0.997 $\pm$ 0.001          | **1.000 $\pm$ 0.000** | **0.914 $\pm$ 0.000** | 19.913 $\pm$ 0.091          | 0.084 $\pm$ 0.000          |
> >
> > **Table 3: Unconditional Generation Results on Guacamol Dataset**
> >  | Methods | Valid $\uparrow$           | Unique$\uparrow$           | Novelty$\uparrow$          | Diversity$\uparrow$        | FCD$\downarrow$             | NSPDK MMD$\downarrow$      |
> > |-----------------|------------------------------------|------------------------------------|------------------------------------|------------------------------------|-------------------------------------|------------------------------------|
> > | GDSS    | **1.000 $\pm$ 0.000** | 0.986 $\pm$ 0.001          | 0.996 $\pm$ 0.001          | 0.892 $\pm$ 0.000          | 40.291 $\pm$ 0.072          | 0.058 $\pm$ 0.000          |
> > | DiGress | 0.875 $\pm$ 0.005          | **1.000 $\pm$ 0.000** | **0.999 $\pm$ 0.001** | 0.904 $\pm$ 0.000          | **12.069 $\pm$ 0.051** | **0.018 $\pm$ 0.000** |
> >  | PS-VAE  | **1.000 $\pm$ 0.000** | 0.998 $\pm$ 0.000          | 0.998 $\pm$ 0.000          | **0.905 $\pm$ 0.000** | 24.105 $\pm$ 0.082          | 0.090 $\pm$ 0.000          |
> >  | GLDM    | **1.000 $\pm$ 0.000** | 0.998 $\pm$ 0.000          | 0.998 $\pm$ 0.000          | 0.904 $\pm$ 0.000          | 23.879 $\pm$ 0.041          | 0.095 $\pm$ 0.000          |
> > | HGLDM   | **1.000 $\pm$ 0.000** | 0.999 $\pm$ 0.001          | **0.999 $\pm$ 0.000** | **0.905 $\pm$ 0.000** | 23.845 $\pm$ 0.098          | 0.095 $\pm$ 0.001          |

---

> > > ### Author Response · Authors · 2023-11-21
> > >
> > > **Table 4: Unconditional Generation Results on MOSES Dataset**
> > > | Methods | Valid $\uparrow$           | Unique$\uparrow$           | Novelty$\uparrow$          | Diversity$\uparrow$        | FCD$\downarrow$             | NSPDK MMD$\downarrow$      |
> > > |-----------------|------------------------------------|------------------------------------|------------------------------------|------------------------------------|-------------------------------------|------------------------------------|
> > >  | GDSS    | **1.000 $\pm$ 0.000** | 0.994 $\pm$ 0.003          | 0.999 $\pm$ 0.000          | 0.899 $\pm$ 0.000          | 21.265 $\pm$ 0.249          | 0.037 $\pm$ 0.005          |
> > > | DiGress | 0.858 $\pm$ 0.005          | **1.000 $\pm$ 0.000** | 0.996 $\pm$ 0.001          | 0.886 $\pm$ 0.000          | **9.228 $\pm$ 0.081**  | **0.010 $\pm$ 0.000** |
> > >  | PS-VAE  | **1.000 $\pm$ 0.000** | 0.999 $\pm$ 0.000          | **1.000 $\pm$ 0.000** | 0.905 $\pm$ 0.000          | 26.401 $\pm$ 0.078          | 0.079 $\pm$ 0.000          |
> > >  | GLDM    | **1.000 $\pm$ 0.000** | 0.998 $\pm$ 0.000          | **1.000 $\pm$ 0.000** | 0.905 $\pm$ 0.000          | 26.365 $\pm$ 0.095          | 0.077 $\pm$ 0.001          |
> > >  | HGLDM   | **1.000 $\pm$ 0.000** | 0.999 $\pm$ 0.000          | **1.000 $\pm$ 0.000** | **0.906 $\pm$ 0.000** | 25.815 $\pm$ 0.053          | 0.072 $\pm$ 0.000          |
> > >
> > >
> > > **W4. The efficiency of the proposed methods seems to come from the light model architecture with smaller dimensions than the model architecture used in the DiGress.**
> > >
> > > R4: We argue that obtaining comparable results to DiGress with **fewer parameters is also a demonstration of efficiency. Actually, this is one of the advantages of our model**. Furthermore, we can see that our model has only slightly over half the number of parameters compared to DiGress, yet it trains 3.6 times faster and samples 83.5 times faster.
> > >
> > > ---
> > > ### Questions:
> > > **Q1. In Section 4.1.1, to my understanding, the number of subgraph embeddings should not be the number of nodes. If so, how do you sample the number of subgraph embeddings at the beginning of the sampling stage?**
> > >
> > > R1: Because there is no node and subgraph alignment issue in the Decoder of PSVAE, during diffusion training, we replicate the subgraph embedding across all nodes within the subgraph to form the subgraph latent embedding. This ensures that the subgraph latent embedding has the same dimension as the node-level embedding. During sampling, we sample subgraph embeddings from the same dimension and match them to the embeddings of the subgraphs where each node belongs.
> > >
> > > **Q2. How do you get the subgraph embeddings from the PS-VAE architecture?**
> > >
> > > R2: We have followed the approach of PSVAE. First, we use the principal subgraph extraction algorithm in PS-VAE to construct the subgraph vocabulary. Then, we obtain the subgraph embeddings through the embedding layer of the encoder in PS-VAE.
> > >
> > > **Q3. In Table 3, how did you measure the mean absolute error (MAE) on the unconditional setting? Does it mean training on the selected 100 molecules without the conditions?**
> > >
> > > R3: Yes, we train the unconditional generation structure based on the selected molecues and measure the MAE between the generated molecules and the selected molecules.
> > >
> > > **Q4. Why are some reported values different from the original papers? For example, the novelty and uniqueness of PS-VAE on the QM9 dataset and the validity of GDSS on the ZINC250k dataset.**
> > >
> > > R4: For the PS-VAE model, we found that lowering the weight of the KL loss can lead to better results than those reported in the original paper. We explain this phenomenon in Appendix B. The reason for the differences in the GDSS results is that we reproduced the GDSS code and discovered that we achieved better  performances. As a result, we reported the improved results.

---

> > > > ### Comment · Reviewer_vEbk · 2023-11-22
> > > >
> > > > I appreciate the detailed explanation and additional experiments. However, I am still concerned about the experimental settings and the contribution of this work.
> > > >
> > > > 1. Regarding the validity check :
> > > > To the best of my knowledge, PS-VAE actually performs the validity check while adding the bond between fragments as described in Algorithm 3 of the PS-VAE's paper. Did you change your decoding stage from the one of PS-VAE?
> > > >
> > > > 2. Regarding the additional experiments with FCD and NSPDK metrics:
> > > > I am afraid that the proposed method is not capable of learning the practical molecular properties because of their inferior performance on FCD and NSPDK metrics as shown in the additional experimental results.
> > > > FCD and NSPDK are related to biological, chemical, and structural properties since FCD takes into account the chemically and biologically relevant information and NSPDK considers the structural connectivity based on their atom types, whereas QED and logP are easily optimizable [1]. I believe that the low FCD and NSPDK MMD values demonstrate the capability of capturing the molecular properties within the training dataset, which can lead to practical utilization. However, the proposed method does not show the better performance of FCD and NSPDK, and thus it seems not to contribute a lot to the graph generation field.
> > > >
> > > > 3. Regarding the contribution:
> > > > Although I agree that generating the node, subgraph, and graph embeddings can reduce the computational cost, the generation quality of the proposed method is not clearly demonstrated, which does not seem to contribute to the graph generation field. To clearly convince the readers of the contribution of this work, I suggest conducting another goal-directed benchmark that is closely related to the actual application such as binding affinity as in MOOD [2].
> > > >
> > > >
> > > > [1] Nathan Brown et al., "GuacaMol: Benchmarking Models for de Novo Molecular Design", JCIM, 2019.
> > > >
> > > > [2] Seul Lee et al., "Exploring Chemical Space with Score-based Out-of-distribution Generation", ICML, 2023.

---

### Official Review · Reviewer_NAaX · 2023-10-30

**Soundness:** 3 good
**Presentation:** 3 good
**Contribution:** 3 good
**Rating:** 8
**Confidence:** 4

**Summary:**

The paper introduces a form of latent diffusion ala Rombach et al. for graphs, in particular, molecule generation. For this they combine a PS-VAE autoencoder with a DDIM style denoising diffusion model which leverages a hierarchy-aware GNN which uses a GAT style subgraph embedding update and PNA pooling for the graph embedding update at every layer.  The method is compared against it's constitutent components and two SotA Diffusion baselines (Digress/GDSS) as well as VAE and other methods on QM9,ZINC250K and Guacamol.

**Strengths:**

Overall a good "Nothing to complain about" paper.

Originality:

Latent diffusion for graphs was a thing waiting to be done, but it is still worth doing. The hierarchical block is a nice construction, as is the continuous-discrete combo.

Quality: The evaluation including Guacamol is good, the QM9 and ZINC250k benchmarks show impressive results.
clarity: The paper is very clearly presented and the appendix, while sparse, gives most of the information required for presenting things.
Significance: Getting hierarchical graph modeling like this going is likely to have a very high impapct, iff the method generalizes.

**Weaknesses:**

- I'd like to see error bars indicating variance accross multiple seeds  if possible
- While QM9 and ZINC250k performance is imprressive, these graphs are kind of solved. What is the performance on MOSES or shapenet?

**Questions:**

1. To clarify, you are evaluating all datasets without hydrogens?
2. The PS-VAE is not permutation equivariant right (or does it canonicalize things)? Did you do any experiments with a purely equivariant backbone?
3. Purely because I found this [paper today](https://arxiv.org/abs/2210.02410) and found the idea exciting, if you manage to perform any diversity quantification using graph embedding similarity across the datasets, I'd be curious how the models differ. This is purely a nerd sharing a neat idea though, not a critique of the paper.
4.  There is no limitations section which is always sus, are there really *no* downsides and limitations worth discussing?

---

> ### Author Response · Authors · 2023-11-21
> **Response to Reviewer NAaX**
>
> We greatly appreciate the reviewer's recognition of the soundness and contributions of our work. In response to the reviewer's request, we included the error bars indicating variance accross multiple seeds, as well as evaluation results on MOSES dataset. Additionally, we added a new metric, diversity, and the Limitation Section in the revised version.
>
> ---
> ### Weaknesses:
> **W1. I'd like to see error bars indicating variance accross multiple seeds if possible.**
>
>  R1: Thank you for your suggestion. During the rebuttal period, we have conducted eight samplings with  different seeds and reported the mean and variance in the revised version. The new results are shown as below:
>
>  **Table 1: Unconditional Generation Results on QM9 Dataset**
>  Methods | Valid $\uparrow$           | Unique$\uparrow$           | Novelty$\uparrow$          | Diversity$\uparrow$        | FCD$\downarrow$             | NSPDK MMD$\downarrow$      |
> |-----------------|------------------------------------|------------------------------------|------------------------------------|------------------------------------|-------------------------------------|------------------------------------|
> | GDSS    | 0.957 $\pm$ 0.000          | 0.982 $\pm$ 0.003          | 0.988 $\pm$ 0.001          | **0.925 $\pm$ 0.000** | 2.959 $\pm$ 0.040           | 0.003 $\pm$ 0.000          |
> | DiGress | 0.992 $\pm$ 0.003          | 0.960 $\pm$ 0.001          | 0.391 $\pm$ 0.001          | 0.920 $\pm$ 0.000          | **2.123 $\pm$ 0.033**  | **0.001 $\pm$ 0.000** |
> | PS-VAE  | **1.000 $\pm$ 0.000** | 0.981 $\pm$ 0.002          | 0.996 $\pm$ 0.000          | 0.881 $\pm$ 0.000          | 16.877 $\pm$ 0.059          | 0.059 $\pm$ 0.000          |
> | GLDM    | **1.000 $\pm$ 0.000** | 0.982 $\pm$ 0.001          | 0.996 $\pm$ 0.000          | 0.884 $\pm$ 0.000          | 14.829 $\pm$ 0.169          | 0.050 $\pm$ 0.001          |
> | HGLDM   | **1.000 $\pm$ 0.000** | **0.985 $\pm$ 0.002** | **0.997 $\pm$ 0.000** | 0.884 $\pm$ 0.000          | 14.576 $\pm$ 0.183          | 0.047 $\pm$ 0.001          |
>
> **Table 2: Unconditional Generation Results on ZINC250K Dataset**
> | Methods | Valid $\uparrow$           | Unique$\uparrow$           | Novelty$\uparrow$          | Diversity$\uparrow$        | FCD$\downarrow$             | NSPDK MMD$\downarrow$      |
> |-----------------|------------------------------------|------------------------------------|------------------------------------|------------------------------------|-------------------------------------|------------------------------------|
> | GDSS    | **1.000 $\pm$ 0.000** | 0.997 $\pm$ 0.001          | **1.000 $\pm$ 0.000** | 0.902 $\pm$ 0.000          | 16.086 $\pm$ 0.071          | **0.018 $\pm$ 0.000** |
> | DiGress | 0.565 $\pm$ 0.005          | **1.000 $\pm$ 0.000** | **1.000 $\pm$ 0.000** | 0.882 $\pm$ 0.000          | **13.042 $\pm$ 0.164** | 0.031 $\pm$ 0.001          |
>  | PS-VAE  | **1.000 $\pm$ 0.000** | 0.993 $\pm$ 0.001          | **1.000 $\pm$ 0.000** | 0.912 $\pm$ 0.000          | 20.386 $\pm$ 0.061          | 0.085 $\pm$ 0.001          |
> | GLDM    | **1.000 $\pm$ 0.000** | 0.994 $\pm$ 0.001          | **1.000 $\pm$ 0.000** | 0.913 $\pm$ 0.000          | 20.444 $\pm$ 0.088          | 0.086 $\pm$ 0.001          |
>  | HGLDM   | **1.000 $\pm$ 0.000** | 0.997 $\pm$ 0.001          | **1.000 $\pm$ 0.000** | **0.914 $\pm$ 0.000** | 19.913 $\pm$ 0.091          | 0.084 $\pm$ 0.000          |
>
> **Table 3: Unconditional Generation Results on Guacamol Dataset**
>  | Methods | Valid $\uparrow$           | Unique$\uparrow$           | Novelty$\uparrow$          | Diversity$\uparrow$        | FCD$\downarrow$             | NSPDK MMD$\downarrow$      |
> |-----------------|------------------------------------|------------------------------------|------------------------------------|------------------------------------|-------------------------------------|------------------------------------|
> | GDSS    | **1.000 $\pm$ 0.000** | 0.986 $\pm$ 0.001          | 0.996 $\pm$ 0.001          | 0.892 $\pm$ 0.000          | 40.291 $\pm$ 0.072          | 0.058 $\pm$ 0.000          |
> | DiGress | 0.875 $\pm$ 0.005          | **1.000 $\pm$ 0.000** | **0.999 $\pm$ 0.001** | 0.904 $\pm$ 0.000          | **12.069 $\pm$ 0.051** | **0.018 $\pm$ 0.000** |
>  | PS-VAE  | **1.000 $\pm$ 0.000** | 0.998 $\pm$ 0.000          | 0.998 $\pm$ 0.000          | **0.905 $\pm$ 0.000** | 24.105 $\pm$ 0.082          | 0.090 $\pm$ 0.000          |
>  | GLDM    | **1.000 $\pm$ 0.000** | 0.998 $\pm$ 0.000          | 0.998 $\pm$ 0.000          | 0.904 $\pm$ 0.000          | 23.879 $\pm$ 0.041          | 0.095 $\pm$ 0.000          |
> | HGLDM   | **1.000 $\pm$ 0.000** | 0.999 $\pm$ 0.001          | **0.999 $\pm$ 0.000** | **0.905 $\pm$ 0.000** | 23.845 $\pm$ 0.098          | 0.095 $\pm$ 0.001          |

---

> > ### Author Response · Authors · 2023-11-21
> >
> > **Table 4: Unconditional Generation Results on MOSES Dataset**
> > | Methods | Valid $\uparrow$           | Unique$\uparrow$           | Novelty$\uparrow$          | Diversity$\uparrow$        | FCD$\downarrow$             | NSPDK MMD$\downarrow$      |
> > |-----------------|------------------------------------|------------------------------------|------------------------------------|------------------------------------|-------------------------------------|------------------------------------|
> >  | GDSS    | **1.000 $\pm$ 0.000** | 0.994 $\pm$ 0.003          | 0.999 $\pm$ 0.000          | 0.899 $\pm$ 0.000          | 21.265 $\pm$ 0.249          | 0.037 $\pm$ 0.005          |
> > | DiGress | 0.858 $\pm$ 0.005          | **1.000 $\pm$ 0.000** | 0.996 $\pm$ 0.001          | 0.886 $\pm$ 0.000          | **9.228 $\pm$ 0.081**  | **0.010 $\pm$ 0.000** |
> >  | PS-VAE  | **1.000 $\pm$ 0.000** | 0.999 $\pm$ 0.000          | **1.000 $\pm$ 0.000** | 0.905 $\pm$ 0.000          | 26.401 $\pm$ 0.078          | 0.079 $\pm$ 0.000          |
> >  | GLDM    | **1.000 $\pm$ 0.000** | 0.998 $\pm$ 0.000          | **1.000 $\pm$ 0.000** | 0.905 $\pm$ 0.000          | 26.365 $\pm$ 0.095          | 0.077 $\pm$ 0.001          |
> >  | HGLDM   | **1.000 $\pm$ 0.000** | 0.999 $\pm$ 0.000          | **1.000 $\pm$ 0.000** | **0.906 $\pm$ 0.000** | 25.815 $\pm$ 0.053          | 0.072 $\pm$ 0.000          |
> >
> > **W2. While QM9 and ZINC250k performance is imprressive, these graphs are kind of solved. What is the performance on MOSES or shapenet?**
> >
> > R2: Thank you for your suggestion. During the rebuttal period, we have added experimental results on the MOSES dataset. The new results are shown in Table 4. Our proposed model achieved better Valid, Novelty, Diversity and comparable Unique scores compared to the baseline methods on the MOSES dataset. In the newly added FCD and NSPDK metrics, although our proposed HGLDM does not outperform diffusion-based methods, it shows further improvement compared to the backbone model PSVAE. Additionally, we believe that molecular generation tasks, which focus more on task-specific molecular properties, have greater practical value than general graph generation tasks that primarily emphasize learning graph distributions. Therefore, during the rebuttal period, we included a latent embedding interpolation experiment to further illustrate the advantages of the latent diffusion model framework in molecular generation tasks. For more details, please refer to General Response G1 and Section 5.2 in the revised version. In addition, the ShapeNet dataset we retrieved is a 3D point cloud data, while our work mainly focuses on 2D molecular generation, so it is not applicable for us to use this dataset.
> >
> > ---
> > ### Questions:
> > **Q1. To clarify, you are evaluating all datasets without hydrogens?**
> >
> > R1: Yes, we removed the hydrogens.
> >
> > **Q2. The PS-VAE is not permutation equivariant right (or does it canonicalize things)? Did you do any experiments with a purely equivariant backbone?**
> >
> > R2: The encoder of PSVAE is permutation equivariant, but the decoder is not, as its output is SMILES strings that only need to be canonicalized. The decoder of PSVAE proposes a two-step subgraph assembling strategy, which first sequentially predicts a set of fragments and then globally assembles all generated subgraphs. This two-step approach makes it less dependent on permutations compared to other traditional VAE-based models. As far as we know, we have not yet found a suitable decoder that is permutation equivariant.
> >
> > **Q3. Purely because I found this paper [1] today and found the idea exciting, if you manage to perform any diversity quantification using graph embedding similarity across the datasets, I'd be curious how the models differ. This is purely a nerd sharing a neat idea though, not a critique of the paper.**
> > [1] https://arxiv.org/abs/2210.02410
> >
> > R3: Thank you for your suggestion. Because baseline method DiGress, which is based on Discrete Diffusion, does not containt graph embeddings, the metric described in [1] cannot be applied. Instead, we have supplemented our evaluation with another commonly used metric in the molecular generation task, Diversity [2]. Diversity evaluates the internal diversity of a set of molecules, defined as the average pairwise Tanimoto distance between the Morgan fingerprints. Our model demonstrates better diversity compared to the baseline methods across most datasets. The new results are shown in Tables 1-4.
> > [2] Kexin Huang, Tianfan Fu, Wenhao Gao, Yue Zhao, Yusuf Roohani, Jure Leskovec, Connor W Coley, Cao Xiao, Jimeng Sun, and Marinka Zitnik. Therapeutics data commons: Machine learning datasets and tasks for drug discovery and development. Proceedings of Neural Information Processing Systems, NeurIPS Datasets and Benchmarks, 2021

---

> > > ### Author Response · Authors · 2023-11-21
> > >
> > > **Q4. There is no limitations section which is always sus, are there really no downsides and limitations worth discussing?**
> > >
> > > R4: Thank you for your suggestion. We have added the Limitations Section in the appendix of the revised version due to the page limitation.

---

> > > ### Comment · Reviewer_NAaX · 2023-11-21
> > >
> > > Thank you for the comments and additional evaluation. One slight pushback: The shapenet ablation would have been interesting for non-molecular data (and presumably, different latent factors). Molecules are also 3D objects.

---

> > > > ### Author Response · Authors · 2023-11-21
> > > > **Thank you for your response**
> > > >
> > > > Dear reviewer NAaX,
> > > >
> > > > Thank you for your response and support. We will complete the training of multiple models as soon as possible and update the latest results in the final version of the paper.
> > > >
> > > > Regarding the Shapenet dataset, we would like to clarify that our goal is to propose a model for generating 2D molecular graphs, and therefore we did not consider 3D molecular data with position information. Transferring a 2D generation model to 3D object generation is not trivial, as we need to address issues such as model equivariance. Thank you for your suggestion. We will discuss the potential application of our model to the Shapenet dataset in the Related Work Section, and plan to implement our model to be compatible with 3D objects in our future work.

---

> > ### Comment · Reviewer_NAaX · 2023-11-21
> >
> > Just to clarify: my comment was about training multiple models, not multiple samplings. All of these numbers come from a single model?

---

### Official Review · Reviewer_bWg3 · 2023-11-01

**Soundness:** 2 fair
**Presentation:** 3 good
**Contribution:** 2 fair
**Rating:** 5
**Confidence:** 4

**Summary:**

The submission proposes a latent diffusion model for graph generation. The problem definition is very common and can be treated as a distribution-fitting problem. The framework utilizes PS-VAE as an encoding-decoding model. And apply a diffusion model over latent variables.

**Strengths:**

1. The presentation is good.
2. The structure design for the diffusion model is reasonable.

**Weaknesses:**

1. Equation (2) is not correct. The probability for each node is computed twice. I think the correct definition should be $\prod p(x_i) \prod \prod (e_{ij})$.

2. The submission claims that they first introduce the latent diffusion model into graph generation. This overclaims the contributions. [1] and many other previous works use latent diffusion models for graph generation tasks. I think the basic idea is the same: only diffuse node variables, and decode edge types from them. This is very common in the area.

3. It is not reasonable to design a hierarchical diffusion model. There is no need to sample three variables $z^x, z^M, z^G$ at the same time. As a hierarchical model, the decoding process of PS-VAE is $ G \sim q(G|z^M)q(z^M|z^G)$. That is, decoding a subgraph from a graph-level vector by a GRU, and then predicting the connection for the subgraphs. So actually, we only need to define a diffusion model over graph-level $z^G$. During the sampling process, we first sample $z^G$ from the diffusion model, and then use decoding of PS-VAE to get the graph. The current framework actually learns the decoding part twice, during the training of PS-VAE, the relationship between each level has been learned already. However, the diffusion model learns it one more time.

4. The results lack MMD metrics. I think it is very important to check the distribution of the graphs.

[1] https://arxiv.org/pdf/2211.10794.pdf

**Questions:**

1. "However, these approaches sacrifice the random exploration ability to ensure that the final noisy data conforms to the appropriate discrete category distribution. " Why do you make such claims? The definition for the distribution of the discrete variables is different. And people can also define a discrete diffusion process over them such as [1] and many other works. The performance is also very good and I think people should select models based on the specific problem. There is no any conclusion to support that continuous features is better than discrete process.

[1]https://arxiv.org/pdf/2209.14734.pdf

---

> ### Author Response · Authors · 2023-11-21
> **Response to Reviewer bWg3**
>
> We thank the reviewer for the provided corrections and suggestions. After reading the comments in the weaknesses 1, 2, and 3 seriously, we are afraid that the reviewer has probably misunderstood the contributions and the model design of our paper. We will try our best to eliminate the misunderstandings via the following responses, and sincerely hope that the reviewer raises any further questions if our responses are still confused.
>
> ---
> ### Weaknesses:
> **W1. Equation (2) is not correct. The probability for each node is computed twice. I think the correct definition should be $\prod p(x_i)\prod\prod(e_{ij})$.**
>
> R1: Thank you for your correction. We have revised it to the correct form in the revised version.
>
> **W2. The submission claims that they first introduce the latent diffusion model into graph generation. This overclaims the contributions. [1] and many other previous works use latent diffusion models for graph generation tasks. I think the basic idea is the same: only diffuse node variables, and decode edge types from them. This is very common in the area.**
> [1] https://arxiv.org/pdf/2211.10794.pdf
>
> R2: Sorry for the misunderstanding here. In the abstract, the word 'first' is meant to convey that we 'first' propose GLDM in this paper and then propose HGLDM. As proposed in the Introduction Section, **our main contribution is that we incorporate the latent diffusion framework and hierarchical structure to address the suboptimal diffusion results caused by using only graph-level embeddings**. Although latent diffusion has been adopted for graph generation in some related work, they only perform diffusion at the node level [1] and ignore the hierarchical information. In the revised version, we have modified the abstract to avoid the misunderstanding and included [1] in the Related Work section.

---

> > ### Author Response · Authors · 2023-11-21
> >
> > **W3. It is not reasonable to design a hierarchical diffusion model. There is no need to sample three variables $z^x$, $z^M$, $z^G$ at the same time. As a hierarchical model, the decoding process of PS-VAE is $G~q(G|z^M)q(z^M|z^G)$. That is, decoding a subgraph from a graph-level vector by a GRU, and then predicting the connection for the subgraphs. So actually, we only need to define a diffusion model over graph-level $z^G$. During the sampling process, we first sample $z^G$ from the diffusion model, and then use decoding of PS-VAE to get the graph. The current framework actually learns the decoding part twice, during the training of PS-VAE, the relationship between each level has been learned already. However, the diffusion model learns it one more time.**
> >
> > R3: Sorry for the misunderstanding, we need to make some clarifications here.
> >
> > **Clarification 1**: **The idea of "we first sampling $z^G$ from the diffusion model, and then using decoding of PS-VAE to get the graph" is actually the GLDM model that we proposed in this paper.** However, due to its unsatisfactory experimental results, we further propose the hierarchical model HGLDM to capture the node-level and structural information. Our experimental results demonstrate the improvement of introducing node-level and subgraph-level embeddings on generating graph embeddings. Furthermore, we argue that during the training of PS-VAE, the graph level embedding is obtained through sum pooling, the relationships between each level have not been well learned. **Through the hierarchical denoising neural network proposed in this paer, we learn the relationships between each level to ensure that the latent embeddings generated through diffusion contains more node-level and subgraph information than the graph latent embedding encoded by PS-VAE.** During the rebuttal period, we conducted a new experiment on the interpolation of latent embeddings, which further demonstrates the enhancement of the HGLDM over PSVAE. we randomly sample two latent embeddings, $\mathbf{z}^{G_1}$ and $\mathbf{z}^{G_2}$, and generate the interpolated embeddings $G'$ through $\mathbf{z}^{G'}=\lambda\mathbf{z}^{G_1}+(1-\lambda)\mathbf{z}^{G_2}$, where $\lambda \in (0, 1)$. Based on the visualized results of these latent embeddings (Figure 4 in the revised version), we can draw two key conclusions. Firstly, the clear boundaries between embeddings of different properties in the scatter plot, along with the high property values retained by the interpolated embeddings, suggest that **latent embeddings can effectively distinguish between molecules with varying property values**. This provides a solid foundation for further conditional generation using these latent embeddings. Secondly, **latent embeddings trained by HGLDM capture more distinct features of different property values compared to PS-VAE latent embeddings**, as evidenced by the farther distances between HGLDM latent embeddings with different property values and the higher property values of molecules decoded from HGLDM latent embeddings interpolation. This highlights the enhancement of the latent diffusion framework over VAE-based models (Please refer to General Response G1 and Section 5.2 in the revised version  for more details).
> >
> > **Clarification 2**: "The current framework actually learns the decoding part twice" is incorrect. As described in Section 4.3 and Algorithm 1 in the Appendix, **the decoder is only trained once**. When training the diffusion model, we only predict the noise added to the latent embedding and do not need to train the decoder.
> >
> > **W4. The results lack MMD metrics. I think it is very important to check the distribution of the graphs.**
> >
> > R4: Thank you for your suggestion. We have included the NSPDK MMD metric in the revised version. NSPDK (Neighborhood subgraph pairwise distance kernel) MMD (Costa & De Grave, 2010) is the MMD between the generated molecules and test molecules which takes into account the node and edge features for evaluation. The new evaluation results are shown below. Although the proposed method does not achieve best results in NSPDK MMD, this is mainly limited by the VAE backbone model, and we have improved upon PSVAE in terms of  NSPDK MMD across all the datasets. **Furthermore, unlike general graph generation that primarily focuses on learning graph distributions, we believe that comparing distribution similarity in molecular generation tasks without considering molecular properties lacks practical significance**. Therefore, after verifying that the performance of the prpoposed method in terms of metrics such as validity and novelty are comparable, indicating that our model performs well in unconditional generation, we shift our focus to conditional molecular generation  that can incorporate more conditional information into the generation process.

---

> > > ### Author Response · Authors · 2023-11-21
> > >
> > > **Table 1: Unconditional Generation Results on QM9 Dataset**
> > >  Methods | Valid $\uparrow$           | Unique$\uparrow$           | Novelty$\uparrow$          | Diversity$\uparrow$        | FCD$\downarrow$             | NSPDK MMD$\downarrow$      |
> > > |-----------------|------------------------------------|------------------------------------|------------------------------------|------------------------------------|-------------------------------------|------------------------------------|
> > > | GDSS    | 0.957 $\pm$ 0.000          | 0.982 $\pm$ 0.003          | 0.988 $\pm$ 0.001          | **0.925 $\pm$ 0.000** | 2.959 $\pm$ 0.040           | 0.003 $\pm$ 0.000          |
> > > | DiGress | 0.992 $\pm$ 0.003          | 0.960 $\pm$ 0.001          | 0.391 $\pm$ 0.001          | 0.920 $\pm$ 0.000          | **2.123 $\pm$ 0.033**  | **0.001 $\pm$ 0.000** |
> > > | PS-VAE  | **1.000 $\pm$ 0.000** | 0.981 $\pm$ 0.002          | 0.996 $\pm$ 0.000          | 0.881 $\pm$ 0.000          | 16.877 $\pm$ 0.059          | 0.059 $\pm$ 0.000          |
> > > | GLDM    | **1.000 $\pm$ 0.000** | 0.982 $\pm$ 0.001          | 0.996 $\pm$ 0.000          | 0.884 $\pm$ 0.000          | 14.829 $\pm$ 0.169          | 0.050 $\pm$ 0.001          |
> > > | HGLDM   | **1.000 $\pm$ 0.000** | **0.985 $\pm$ 0.002** | **0.997 $\pm$ 0.000** | 0.884 $\pm$ 0.000          | 14.576 $\pm$ 0.183          | 0.047 $\pm$ 0.001          |
> > >
> > > **Table 2: Unconditional Generation Results on ZINC250K Dataset**
> > > | Methods | Valid $\uparrow$           | Unique$\uparrow$           | Novelty$\uparrow$          | Diversity$\uparrow$        | FCD$\downarrow$             | NSPDK MMD$\downarrow$      |
> > > |-----------------|------------------------------------|------------------------------------|------------------------------------|------------------------------------|-------------------------------------|------------------------------------|
> > > | GDSS    | **1.000 $\pm$ 0.000** | 0.997 $\pm$ 0.001          | **1.000 $\pm$ 0.000** | 0.902 $\pm$ 0.000          | 16.086 $\pm$ 0.071          | **0.018 $\pm$ 0.000** |
> > > | DiGress | 0.565 $\pm$ 0.005          | **1.000 $\pm$ 0.000** | **1.000 $\pm$ 0.000** | 0.882 $\pm$ 0.000          | **13.042 $\pm$ 0.164** | 0.031 $\pm$ 0.001          |
> > >  | PS-VAE  | **1.000 $\pm$ 0.000** | 0.993 $\pm$ 0.001          | **1.000 $\pm$ 0.000** | 0.912 $\pm$ 0.000          | 20.386 $\pm$ 0.061          | 0.085 $\pm$ 0.001          |
> > > | GLDM    | **1.000 $\pm$ 0.000** | 0.994 $\pm$ 0.001          | **1.000 $\pm$ 0.000** | 0.913 $\pm$ 0.000          | 20.444 $\pm$ 0.088          | 0.086 $\pm$ 0.001          |
> > >  | HGLDM   | **1.000 $\pm$ 0.000** | 0.997 $\pm$ 0.001          | **1.000 $\pm$ 0.000** | **0.914 $\pm$ 0.000** | 19.913 $\pm$ 0.091          | 0.084 $\pm$ 0.000          |
> > >
> > > **Table 3: Unconditional Generation Results on Guacamol Dataset**
> > >  | Methods | Valid $\uparrow$           | Unique$\uparrow$           | Novelty$\uparrow$          | Diversity$\uparrow$        | FCD$\downarrow$             | NSPDK MMD$\downarrow$      |
> > > |-----------------|------------------------------------|------------------------------------|------------------------------------|------------------------------------|-------------------------------------|------------------------------------|
> > > | GDSS    | **1.000 $\pm$ 0.000** | 0.986 $\pm$ 0.001          | 0.996 $\pm$ 0.001          | 0.892 $\pm$ 0.000          | 40.291 $\pm$ 0.072          | 0.058 $\pm$ 0.000          |
> > > | DiGress | 0.875 $\pm$ 0.005          | **1.000 $\pm$ 0.000** | **0.999 $\pm$ 0.001** | 0.904 $\pm$ 0.000          | **12.069 $\pm$ 0.051** | **0.018 $\pm$ 0.000** |
> > >  | PS-VAE  | **1.000 $\pm$ 0.000** | 0.998 $\pm$ 0.000          | 0.998 $\pm$ 0.000          | **0.905 $\pm$ 0.000** | 24.105 $\pm$ 0.082          | 0.090 $\pm$ 0.000          |
> > >  | GLDM    | **1.000 $\pm$ 0.000** | 0.998 $\pm$ 0.000          | 0.998 $\pm$ 0.000          | 0.904 $\pm$ 0.000          | 23.879 $\pm$ 0.041          | 0.095 $\pm$ 0.000          |
> > > | HGLDM   | **1.000 $\pm$ 0.000** | 0.999 $\pm$ 0.001          | **0.999 $\pm$ 0.000** | **0.905 $\pm$ 0.000** | 23.845 $\pm$ 0.098          | 0.095 $\pm$ 0.001          |

---

> > > > ### Author Response · Authors · 2023-11-21
> > > >
> > > > **Table 4: Unconditional Generation Results on MOSES Dataset**
> > > > | Methods | Valid $\uparrow$           | Unique$\uparrow$           | Novelty$\uparrow$          | Diversity$\uparrow$        | FCD$\downarrow$             | NSPDK MMD$\downarrow$      |
> > > > |-----------------|------------------------------------|------------------------------------|------------------------------------|------------------------------------|-------------------------------------|------------------------------------|
> > > >  | GDSS    | **1.000 $\pm$ 0.000** | 0.994 $\pm$ 0.003          | 0.999 $\pm$ 0.000          | 0.899 $\pm$ 0.000          | 21.265 $\pm$ 0.249          | 0.037 $\pm$ 0.005          |
> > > > | DiGress | 0.858 $\pm$ 0.005          | **1.000 $\pm$ 0.000** | 0.996 $\pm$ 0.001          | 0.886 $\pm$ 0.000          | **9.228 $\pm$ 0.081**  | **0.010 $\pm$ 0.000** |
> > > >  | PS-VAE  | **1.000 $\pm$ 0.000** | 0.999 $\pm$ 0.000          | **1.000 $\pm$ 0.000** | 0.905 $\pm$ 0.000          | 26.401 $\pm$ 0.078          | 0.079 $\pm$ 0.000          |
> > > >  | GLDM    | **1.000 $\pm$ 0.000** | 0.998 $\pm$ 0.000          | **1.000 $\pm$ 0.000** | 0.905 $\pm$ 0.000          | 26.365 $\pm$ 0.095          | 0.077 $\pm$ 0.001          |
> > > >  | HGLDM   | **1.000 $\pm$ 0.000** | 0.999 $\pm$ 0.000          | **1.000 $\pm$ 0.000** | **0.906 $\pm$ 0.000** | 25.815 $\pm$ 0.053          | 0.072 $\pm$ 0.000          |
> > > >
> > > > ### Questions:
> > > > **Q1. "However, these approaches sacrifice the random exploration ability to ensure that the final noisy data conforms to the appropriate discrete category distribution. " Why do you make such claims? The definition for the distribution of the discrete variables is different. And people can also define a discrete diffusion process over them such as [1] and many other works. The performance is also very good and I think people should select models based on the specific problem. There is no any conclusion to support that continuous features is better than discrete process.**
> > > > [1] https://arxiv.org/pdf/2209.14734.pdf
> > > >
> > > > R1: Thank you for your question. We make this claim based on the discrete diffusion process in [1] you mentioned. It can be seen from Eq. (6) in [1] that their transition matrix is only related to the marginal distribution of the dataset without adding random noise.  Therefore, we claim that such methods sacrifice the ability of random exploration. From the experimental results, we can also observe that when conducting experiments on datasets with simpler data distributions such as QM9, the Novelty of DiGress decreases significantly compared to other diffusion methods. We agree that ''people should select models based on the specific problem, and there is no any conclusion to support that continuous features is better" because both methods perform good in unconditional task. However, **proving that the reverse generation process of a custom discrete diffusion satisfies the inverse process of its diffusion process is not easy, while this is easier to achieve in continuous Gaussain diffusion and can be more easily extended for conditional generation tasks.**

---

> > > > ### Comment · Reviewer_bWg3 · 2023-12-04
> > > > **Thanks for your reply**
> > > >
> > > > I appreciate authors' detailed explanation. I now understand the whole framework clearly. However, I still feel like it lacks the motivation to adopt such hierarchical generation process. From the new experiment results, there is no improvement for FCD and NSPDK compared with GLDM and other baselines. It indicates the current method can not learn the distribution of the dataset well. I encourage the authors to investigate such issue.

---

### Author Response · Authors · 2023-11-21
**General Response: Improvement during the rebuttal period**

We sincerely thank all the reviewers for your efforts to make this work better. We would like to summarize the new experiments we have conducted during the rebuttal period to further highlight our contributions here and then respond to each reviewer.

---
### G1. Interpolation of latent embeddings
To showcase the improvements offered by the latent diffusion framework in molecular generation, we have conducted three sets of interpolation experiments on latent embeddings of trained PSVAE and HGLDM: QED Interpolation, PlogP Interpolation, and Mixed Interpolation. QED Interpolation involves using only latent embeddings with high QED values, while PlogP Interpolation uses only those with high PlogP values. Mixed interpolation combines two types of latent embeddings, those with high QED and those with high PlogP values. In each interpolation, we randomly sample two latent embeddings, $\mathbf{z}^{G_1}$ and $\mathbf{z}^{G_2}$, and generate the interpolated embeddings $G'$ through $\mathbf{z}^{G'}=\lambda\mathbf{z}^{G_1}+(1-\lambda)\mathbf{z}^{G_2}$, where $\lambda \in (0, 1)$. Please refer to the Section 5.2 in the revised version for more detailed experimental setups.

Based on the visualized results of these latent embeddings (Figure 4 in the revised version), we can draw two key conclusions. Firstly, the clear boundaries between embeddings of different properties in the scatter plot, along with the high property values retained by the interpolated embeddings, suggest that **latent embeddings can effectively distinguish between molecules with varying property values**. This provides a solid foundation for further conditional generation using these latent embeddings. Secondly, **latent embeddings trained by HGLDM capture more distinct features of different property values compared to PS-VAE latent embeddings**, as evidenced by the farther distances between HGLDM latent embeddings with different property values and the higher property values of molecules decoded from HGLDM latent embeddings interpolation. This highlights the enhancement of the latent diffusion framework over VAE-based models.

---
### G2. More evaluation metrics, larger dataset and variance
During the rebuttal period, we have supplemented the following three experiments:

**(1) Adding new evaluation metrics**

We have added three additional metrics, Diversity, FCD and NSPDK MMD, for unconditional generation.
- Diversity [1] evaluates the internal diversity of a set of molecules. The internal diversity is defined as the average pairwise Tanimoto distance between the Morgan fingerprints.
- FCD (Frechet ChemNet Distance) [2] evaluates the distance between the training and generated sets using the activations of the penultimate layer of the ChemNet.
- NSPDK (Neighborhood subgraph pairwise distance kernel) MMD [3] is the MMD between the generated molecules and test molecules, which considers both the node and edge features for evaluation.

**(2) Evaluating on larger dataset - MOSES**

We have included the evaluation results on a larger dataset, MOSES [4], with more than 1.9M molecules.

**(3) Calculating the variance**

We have calculated the variance of each model's performance through eight samplings with different seeds.

The updated results of these experiments are presented below (Table 2 in the revised version). We observed that GLDM and HGLDM outperformed diffusion-based methods in terms of diversity on most of the datasets. Although the proposed method does not achieve the best results in FCD and NSPDK MMD, this is mainly limited by the VAE backbone model, and we have improved upon PS-VAE in terms of diversity, FCD and NSPDK across all the datasets. Furthermore, unlike general graph generation which primarily focuses on learning graph distributions, comparing distribution similarity in molecular generation tasks without considering molecular properties lacks practical significance. Therefore, after verifying that the proposed method performs well in terms of metrics such as validity and novelty are comparable in unconditional generation, we shift our focus to conditional molecular generation  that can incorporate more conditional information to generate molecules with practical properties.

---

> ### Author Response · Authors · 2023-11-21
>
> **Table 1: Unconditional Generation Results on QM9 Dataset**
>  Methods | Valid $\uparrow$           | Unique$\uparrow$           | Novelty$\uparrow$          | Diversity$\uparrow$        | FCD$\downarrow$             | NSPDK MMD$\downarrow$      |
> | ----------------- | ------------------------------------|------------------------------------|------------------------------------|------------------------------------|-------------------------------------|------------------------------------|
> | GDSS    | 0.957 $\pm$ 0.000          | 0.982 $\pm$ 0.003          | 0.988 $\pm$ 0.001          | **0.925 $\pm$ 0.000** | 2.959 $\pm$ 0.040           | 0.003 $\pm$ 0.000          |
> | DiGress | 0.992 $\pm$ 0.003          | 0.960 $\pm$ 0.001          | 0.391 $\pm$ 0.001          | 0.920 $\pm$ 0.000          | **2.123 $\pm$ 0.033**  | **0.001 $\pm$ 0.000** |
> | PS-VAE  | **1.000 $\pm$ 0.000** | 0.981 $\pm$ 0.002          | 0.996 $\pm$ 0.000          | 0.881 $\pm$ 0.000          | 16.877 $\pm$ 0.059          | 0.059 $\pm$ 0.000          |
> | GLDM    | **1.000 $\pm$ 0.000** | 0.982 $\pm$ 0.001          | 0.996 $\pm$ 0.000          | 0.884 $\pm$ 0.000          | 14.829 $\pm$ 0.169          | 0.050 $\pm$ 0.001          |
> | HGLDM   | **1.000 $\pm$ 0.000** | **0.985 $\pm$ 0.002** | **0.997 $\pm$ 0.000** | 0.884 $\pm$ 0.000          | 14.576 $\pm$ 0.183          | 0.047 $\pm$ 0.001          |
>
> **Table 2: Unconditional Generation Results on ZINC250K Dataset**
> | Methods | Valid $\uparrow$           | Unique$\uparrow$           | Novelty$\uparrow$          | Diversity$\uparrow$        | FCD$\downarrow$             | NSPDK MMD$\downarrow$      |
> |-----------------|------------------------------------|------------------------------------|------------------------------------|------------------------------------|-------------------------------------|------------------------------------|
> | GDSS    | **1.000 $\pm$ 0.000** | 0.997 $\pm$ 0.001          | **1.000 $\pm$ 0.000** | 0.902 $\pm$ 0.000          | 16.086 $\pm$ 0.071          | **0.018 $\pm$ 0.000** |
> | DiGress | 0.565 $\pm$ 0.005          | **1.000 $\pm$ 0.000** | **1.000 $\pm$ 0.000** | 0.882 $\pm$ 0.000          | **13.042 $\pm$ 0.164** | 0.031 $\pm$ 0.001          |
>  | PS-VAE  | **1.000 $\pm$ 0.000** | 0.993 $\pm$ 0.001          | **1.000 $\pm$ 0.000** | 0.912 $\pm$ 0.000          | 20.386 $\pm$ 0.061          | 0.085 $\pm$ 0.001          |
> | GLDM    | **1.000 $\pm$ 0.000** | 0.994 $\pm$ 0.001          | **1.000 $\pm$ 0.000** | 0.913 $\pm$ 0.000          | 20.444 $\pm$ 0.088          | 0.086 $\pm$ 0.001          |
>  | HGLDM   | **1.000 $\pm$ 0.000** | 0.997 $\pm$ 0.001          | **1.000 $\pm$ 0.000** | **0.914 $\pm$ 0.000** | 19.913 $\pm$ 0.091          | 0.084 $\pm$ 0.000          |
>
> **Table 3: Unconditional Generation Results on Guacamol Dataset**
>  | Methods | Valid $\uparrow$           | Unique$\uparrow$           | Novelty$\uparrow$          | Diversity$\uparrow$        | FCD$\downarrow$             | NSPDK MMD$\downarrow$      |
> |-----------------|------------------------------------|------------------------------------|------------------------------------|------------------------------------|-------------------------------------|------------------------------------|
> | GDSS    | **1.000 $\pm$ 0.000** | 0.986 $\pm$ 0.001          | 0.996 $\pm$ 0.001          | 0.892 $\pm$ 0.000          | 40.291 $\pm$ 0.072          | 0.058 $\pm$ 0.000          |
> | DiGress | 0.875 $\pm$ 0.005          | **1.000 $\pm$ 0.000** | **0.999 $\pm$ 0.001** | 0.904 $\pm$ 0.000          | **12.069 $\pm$ 0.051** | **0.018 $\pm$ 0.000** |
>  | PS-VAE  | **1.000 $\pm$ 0.000** | 0.998 $\pm$ 0.000          | 0.998 $\pm$ 0.000          | **0.905 $\pm$ 0.000** | 24.105 $\pm$ 0.082          | 0.090 $\pm$ 0.000          |
>  | GLDM    | **1.000 $\pm$ 0.000** | 0.998 $\pm$ 0.000          | 0.998 $\pm$ 0.000          | 0.904 $\pm$ 0.000          | 23.879 $\pm$ 0.041          | 0.095 $\pm$ 0.000          |
> | HGLDM   | **1.000 $\pm$ 0.000** | 0.999 $\pm$ 0.001          | **0.999 $\pm$ 0.000** | **0.905 $\pm$ 0.000** | 23.845 $\pm$ 0.098          | 0.095 $\pm$ 0.001          |

---

> > ### Author Response · Authors · 2023-11-21
> >
> > **Table 4: Unconditional Generation Results on MOSES Dataset**
> > | Methods | Valid $\uparrow$           | Unique$\uparrow$           | Novelty$\uparrow$          | Diversity$\uparrow$        | FCD$\downarrow$             | NSPDK MMD$\downarrow$      |
> > |-----------------|------------------------------------|------------------------------------|------------------------------------|------------------------------------|-------------------------------------|------------------------------------|
> >  | GDSS    | **1.000 $\pm$ 0.000** | 0.994 $\pm$ 0.003          | 0.999 $\pm$ 0.000          | 0.899 $\pm$ 0.000          | 21.265 $\pm$ 0.249          | 0.037 $\pm$ 0.005          |
> > | DiGress | 0.858 $\pm$ 0.005          | **1.000 $\pm$ 0.000** | 0.996 $\pm$ 0.001          | 0.886 $\pm$ 0.000          | **9.228 $\pm$ 0.081**  | **0.010 $\pm$ 0.000** |
> >  | PS-VAE  | **1.000 $\pm$ 0.000** | 0.999 $\pm$ 0.000          | **1.000 $\pm$ 0.000** | 0.905 $\pm$ 0.000          | 26.401 $\pm$ 0.078          | 0.079 $\pm$ 0.000          |
> >  | GLDM    | **1.000 $\pm$ 0.000** | 0.998 $\pm$ 0.000          | **1.000 $\pm$ 0.000** | 0.905 $\pm$ 0.000          | 26.365 $\pm$ 0.095          | 0.077 $\pm$ 0.001          |
> >  | HGLDM   | **1.000 $\pm$ 0.000** | 0.999 $\pm$ 0.000          | **1.000 $\pm$ 0.000** | **0.906 $\pm$ 0.000** | 25.815 $\pm$ 0.053          | 0.072 $\pm$ 0.000          |
> >
> >
> > [1] Kexin Huang, Tianfan Fu, Wenhao Gao, Yue Zhao, Yusuf Roohani, Jure Leskovec, Connor W Coley, Cao Xiao, Jimeng Sun, and Marinka Zitnik. Therapeutics data commons: Machine learning datasets and tasks for drug discovery and development. Proceedings of Neural Information Processing Systems, NeurIPS Datasets and Benchmarks, 2021.
> >
> > [2] Kristina Preuer, Philipp Renz, Thomas Unterthiner, Sepp Hochreiter, and Gunter Klambauer. Fr ́echet chemnet distance: a metric for generative models for molecules in drug discovery. Journal of chemical information and modeling, 58(9):1736–1741, 2018.
> >
> > [3] Fabrizio Costa and Kurt De Grave. Fast neighborhood subgraph pairwise distance kernel. In Proceedings of the 26th International Conference on Machine Learning, pp. 255–262. Omnipress; Madison, WI, USA, 2010.
> >
> > [4] Daniil Polykovskiy, Alexander Zhebrak, Benjamin Sanchez-Lengeling, Sergey Golovanov, Oktai Tatanov, Stanislav Belyaev, Rauf Kurbanov, Aleksey Artamonov, Vladimir Aladinskiy, Mark Veselov, et al. Molecular sets (moses): a benchmarking platform for molecular generation models. Frontiers in pharmacology, 11:565644, 2020.

---

### Meta-Review · Area_Chair_6ZMb · 2023-12-08

**Metareview:**

Summary:

This paper proposes a latent diffusion model that aims to generate hierarchical levels of latent variables such as node-level, subgraph-level, and graph-level, simultaneously. To construct the latent space, the authors leverage the PS-VAE where the decoder converts the graph-level latent variables to the molecule by sequentially predicting the fragments and then predicting the links between fragments. To generate the latent variables of node-, subgraph-, and graph-level, the authors leverage the generative process of DDPM. The authors propose an architecture that models the dependency between node-, subgraph-, and graph-level embeddings alleviating the burden of considering the edge features. The proposed method is evaluated on the molecule generation tasks in the conditional and unconditional settings. The method is compared against it's constitutent components and two SotA Diffusion baselines (Digress/GDSS) as well as VAE and other methods on QM9,ZINC250K and Guacamol.

Strengths:

- Latent diffusion for graphs was a thing waiting to be done, but it is still worth doing.
- The hierarchical block is a nice construction, as is the continuous-discrete combo.
- The evaluation including Guacamol is good, the QM9 and ZINC250k benchmarks show impressive results.
- The paper is very clearly presented and the appendix, while sparse, gives most of the information required for presenting things.
- Getting hierarchical graph modeling like this going is likely to have a very high impapct, iff the method generalizes.
- The auxiliary generation of node and subgraph embeddings can enrich the forwarded information of the diffusion process.
- The paper is well-written and easy to follow.
- The motivation for using the latent approach is clear.
- Using hierarchical embeddings of graphs for graph latent diffusion is novel and shows improvements in conditional molecule generation tasks.

Weaknesses:

- The submission claims that they first introduce the latent diffusion model into graph generation. This overclaims the contributions.
- Many other previous works use latent diffusion models for graph generation tasks.
- It may not be reasonable to design a hierarchical diffusion model. There is no need to sample three variables at the same time.
- During the sampling process, we first sample from the diffusion model, and then use decoding of PS-VAE to get the graph.
- The current framework learns the decoding part twice, during the training of PS-VAE, the relationship between each level has been learned already. However, the diffusion model learns it one more time.
- The results lack MMD metrics. I think it is very important to check the distribution of the graphs.
- Error bars indicating variance accross multiple seeds needed.
- While QM9 and ZINC250k performance is imprressive, these graphs are kind of solved.
- Not clear why the authors select a way to generate graph embeddings and then decode them. Accessing the graph embedding could contain less information than accessing the subgraph- or node- and edge-level latent variables.
- The name of the proposed method is misleading.
- The authors report the validity, uniqueness, and novelty as the main results. However, these metrics seem to be restricted to only measure the sample diversity.
- To demonstrate whether the generative model precisely learns the data distribution, reporting the uniqueness and the novelty is not enough.
- The provided experiments are limited to datasets (e.g., GuacaMol) in which previous diffusion models (e.g., GDSS and DiGress) are applicable. In order to justify the scalability of the proposed method, it should be evaluated in a larger dataset.
- The experimental setting for evaluating the computational efficiency is not clear.
- Generation performance on unconditional molecule generation tasks should be evaluated with more descriptive metrics, for example, FCD, Scaffold similarity, and Fragment similarity.
- Reported metrics, i.e., validity, uniqueness, and novelty fail to measure how similar (e.g., chemical aspects) are the generated molecules to the molecules from the test set
- The quantitative results of Tables 2 and 3 show that the performances of GLDM and HGLDM on unconditional generation tasks are almost the same, whereas there is a significant improvement using the hierarchical approach for conditional generation tasks.
- The performance of GLDM (and HGLDM) comes from the effectiveness of using a latent representation of graphs compared to previous graph diffusion models, not from the diffusion processes. Thereby, analysis of the latent representation, e.g., interpolation in the latent space or clustering of the latent points with respect to certain conditions, would greatly strengthen this work.

Recommendation:

A majority of reviewers lean towards rejection. I, therefore, recommend rejecting the paper and encourage the authors to use the feedback provided to improve the paper and resubmit to another venue.

**Justification For Why Not Higher Score:**

A majority of reviewers lean towards rejection. The reviewers point out many weaknesses in the submitted paper.

**Justification For Why Not Lower Score:**

N/A

---

### Decision · Program_Chairs · 2024-01-16

Reject